# SlimVar for rapid in vivo single-molecule tracking of chromatin regulators in plants

Alex L. Payne-Dwyer [1,2], Geng-Jen Jang [3], Caroline Dean [3] & Mark C. Leake [1,2] ✉

Epigenetic regulation occurs over many rounds of cell division in higher organisms. However, visualisation of the regulators in vivo is limited by imaging dynamic molecules deep in tissue. We report a technology—Variable-angle Slimfield microscopy (SlimVar)—that enables tracking of single fluorescent reporters to 30 μm depth through multiple *Arabidopsis thaliana* root tip cell layers. SlimVar uses rapid photobleaching to resolve tracked particles to molecular steps in intensity. By modifying widefield microscopy to minimise optical aberrations and robustly post-process few-photon signals, SlimVar mitigates performance losses at depth. We use SlimVar to quantify chromatin-protein assemblies in nuclei, finding that two homologous proteins key to epigenetic switching at *FLOWERING LOCUS C* (*FLC*) —cold-induced VERNALISATION INSENSITIVE3 (VIN3) and constitutively expressed VERNALISATION 5 (VRN5)—exhibit dynamic assemblies during *FLC* silencing. Upon cold exposure, the number of assembly molecules increases up to 100% to a median of ~20 molecules. Larger VRN5 assemblies preferentially colocalise with an *FLC lacO* transgenic reporter during prolonged cold and persist after return to warmth. Our findings support a hybrid model of epigenetic memory in which nucleation of histone trimethylation is assisted by dynamic protein assemblies over extended durations. SlimVar offers molecular insights into proteins expressed at physiological levels in tissues.

Understanding the basis of epigenetic silencing remains a major question, with potential implications for both novel medical therapeutics[1] and agricultural biotechnology[2]. It is not yet known how cells process delocalised, long-term sensory information into genetic states that are stable enough to facilitate not only cellular differentiation but also determine key organism-level transitions, including those relevant to medicine such as ageing and disease[3]. Similarly, mysteries remain regarding long-term epigenetic responses to key environmental cues, including seasonal temperature fluctuations, govern the productivity and resilience of crops to challenges such as climate change[4].

A key conserved epigenetic mechanism across eukaryotes, including humans and plants, is Polycomb-mediated silencing[5], which involves modification of trimethylated histone H3 lysine 27 (H3K27me3). H3K27me3 initially accumulates at a nucleation site then spreads across a locus to stably maintain a silenced state through many rounds of cell division[6]. Polycomb Repressive Complex 2 (PRC2) silencing has been well studied in *Arabidopsis* gene *FLOWERING LOCUS C* (*FLC*)[7]. *FLC* encodes a repressor of flowering and is epigenetically silenced during a process called vernalisation by the prolonged cold of winter; lowering repressor levels enables flowering in spring[8–10]. Cold exposure increases the probability that each *FLC* locus will epigenetically switch from ON to OFF states through nucleation of H3K27me3 at an intragenic site[11]. Upon return to warm conditions, H3K27me3 spreads across the locus to give long-term stable silencing[12]. What has been less clear is how the relative stability of the

[1]School of Physics, Engineering and Technology, University of York, York, UK. [2]Department of Biology, University of York, York, UK. [3]Cell and Developmental Biology, John Innes Centre, Norwich Research Park, Norwich, UK. ✉e-mail: mark.leake@york.ac.uk

nucleated state is inherited following cell division. Each nucleation event involves only three nucleosomes, too few to survive random replicative dilution through the classic Polycomb 'read-write' mechanism[13]. Metastable protein assemblies have been proposed to explain the inheritance of the nucleated state[14]. PRC2 accessory proteins are thought to restore the stochastic loss of histone marks by stimulating Polycomb Repressive Complex 2 (PRC2) activity[15]. While individual proteins may only interact transiently with nucleation factors and the locus, a recent model predicts that an assembly with the appropriate positive cooperativity can become dynamically self-sustaining above a threshold number of recruited proteins[14]. In this framework, a sufficiently large assembly of proteins could act as a binary memory element working together with established H3K27me3 machinery at a given locus. This model highlights the number of molecules in the assembly as a key factor.

Two PRC2 accessory proteins required for stable cold-induced silencing at *FLC*, VERNALISATION INSENSITIVE3 (VIN3) and VERNALISATION 5 (VRN5, also known as VIN3-LIKE 1/VIL1), expressed in shoot and root tips are clear candidates for this form of memory storage[16]. VIN3 and VRN5 (collectively called VEL proteins) associate with the PRC2 complex and with the *FLC* nucleation region that accumulates H3K27me3 specifically during cold conditions[17]. At <15 °C, *VIN3* expression gradually rises over several weeks, while in warm conditions >15 °C, expression decays rapidly within ~4 h[8,9]. VIN3 and its assemblies could therefore in principle report the duration of cold conditions during winter to promote an epigenetic switch. However, due to its instability in the warm, one or more additional factors—a promising candidate being VRN5—are required at the *FLC* nucleation region to explain persistent memory of silencing following a change to warm conditions. Both VIN3 and VRN5 contain complex plant homeodomains (PHDs)[16] which do not interact directly with histone tails[18], FNIII domains and C-terminal VEL domains[19]. Of these three domains common to VIN3 and VRN5, the VEL domain mediates head-to-tail interactions. VEL proteins that oligomerise and even form phase-separated droplets under transient overexpression are directly associated with stable silencing at *FLC*[20]. A key question is then: do physiological levels of VIN3 and VRN5 oligomerise sufficiently in the vicinity of *FLC* during vernalisation, to fulfil this model?

However, prior investigation of nuclear oligomers has been constrained by an inability to detect individual, rapidly diffusing protein molecules deep within plant tissue, despite the use of root tips with low autofluorescence and regular, less refractive layers than other plant tissues. Total internal reflection microscopy (TIRF)[21] is not suitable at depth, while traditional epifluorescence, confocal[22], structured illumination[23], lightsheet[24] and Slimfield microscopy[25,26] lack the required combination of sensitivity and speed for single-molecule tracking in live plants, which remains a challenge beyond the first cell layer[27–30]. Although complex, expensive super-resolution methods including lattice lightsheet[31] and MINFLUX[32] are in principle capable of deeper imaging, to date neither has been successfully applied at a molecular scale in plants. In addition, these require both specialised hardware and/or photoactivatable/photoswitchable fluorescent proteins[33] or dyes[34].

Here, we describe development of a photon-efficient imaging technology, Slimfield Variable Angle (SlimVar), which instead uses common fluorescent protein fusions in existing transgenic plants without requiring overexpression, as well as a relatively accessible microscope platform. Adapted from Slimfield microscopy, SlimVar enhances the image contrast using a HILO-like grazing angle of incidence and mitigates optical aberrations at depth to enable dynamic spatial localisation in complex multicellular samples. Single-molecule sensitivity is combined with stepwise photobleaching analysis[25] to quantify the number of molecules in any observed oligomeric assemblies. This sensitivity simplifies the use of transgenic plant lines expressing from as few as a single gene copy and enables imaging of

physiological states of low-abundance nuclear proteins. We detail the principles and operational procedures of SlimVar, and demonstrate its measurement capabilities for tracking single fluorescent protein molecules up to 30 µm deep with lateral spatial precision as fine as 40 nm. We then apply it to rapidly track and quantify VEL proteins in live plant tissue. We find that both VIN3 and VRN5 proteins form assemblies in cell nuclei, composed of consistent dimeric subunits. In lines expressing from single-digit exogenous copies of VIN3 and VRN5, the median assembly comprises up to ~20 molecules of each protein, in agreement with that required for protein memory elements predicted from modelling[14]. We also use an *FLC-lacO*/LacI-YFP transgenic reporter to localise the *FLC* locus and characterise its mobility relative to VEL proteins. Finally, we demonstrate dual-colour SlimVar, which directly shows VRN5 assemblies present at *FLC*. We find that larger VRN5 assemblies preferentially colocalise with *FLC* after long cold exposure and after return to warm conditions, and that this interaction between individual larger oligomers of VRN5 and *FLC* is dynamic on a sub-second timescale.

## Results

### SlimVar enables molecular quantification of diffusing particles deep in tissue

SlimVar is a 2D+ time imaging technique which identifies fluorescent foci from local intensity maxima in each video frame (Fig. 1, Table 1). Foci correspond to at least one, or more generally, a localised group of labelled molecules much smaller than the widefield resolution limit; output parameters include 2D spatial location, total intensity in photons, and signal-to-noise ratio (SNR). If the frame rate is rapid enough to overcome motion blur, foci from sequential frames can be linked into tracks. Since the molecules in a track are spatially correlated, we can infer that they form an intermolecular assembly, and that each track corresponds to one assembly.

Rapid tracking of single diffusive proteins requires minimal background (equivalent to <$10^7$ photons per µm²/s), and a photon-sensitive, low-noise detector (<2 electrons readout per pixel) with millisecond or less sampling. Maximising the emissive output of fluorescent protein tags in these short exposures requires high excitation intensity (~kW/cm²).

Photobleaching dominates under these conditions, which SlimVar uses to capture the emission intensities of both the initial unbleached state of each assembly and its constituent single fluorophores in the same short (~10 s) acquisition.

The ratio of these—stoichiometry—is an estimate of the number of molecules in each assembly. The numerator in this ratio is the track's initial intensity; its relative uncertainty is low since the first of its foci contains many photons. The uncertainty can be reduced further by interpolation from fitting the photobleach trend in time.

The denominator in the ratio relates to single molecule detection events, each containing few photons; there may be very few or no such events that can be associated with a particular assembly, for example if it leaves the detection volume. Instead, we consider the population-average number of photons associated with each fluorophore—the 'characteristic molecular brightness'. It can be determined by averaging the height of individual photobleaching steps[25] (Supplementary Fig. 1). As an average, it is only a reliable estimate for individual assemblies if the intensities are narrowly distributed for fluorophores of the same type in the same environment, and independent of concentration. These conditions are well satisfied by intracellular fluorescent protein fusion constructs, which do not easily self-quench[35]. These intensities represent photons per frame and, if not in a saturating regime for emission, scale with the exposure time, excitation power and the photon collection efficiency of the microscope. However, the characteristic molecular brightness can be internally calibrated within each sample or dataset for a ~20 µm range of working depths. For this reason, it is best practice to acquire additional frames

## Photobleaching analysis workflow

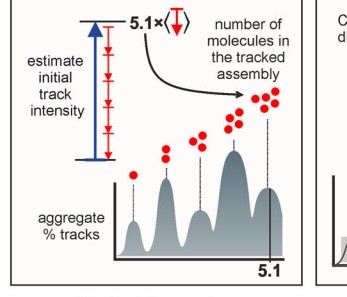

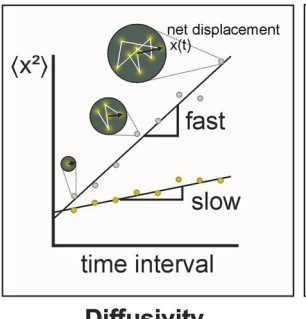

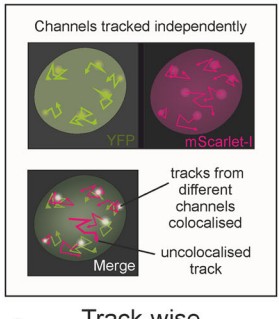

**Fig. 1 | Correlative quantification of diffusing assemblies using SlimVar.** Terms are defined in Table 1. SlimVar delivers **a** rapid photobleaching in image sequences at high (millisecond) framerate, over ~10 s cumulative exposure time *t*, to outpace molecular diffusion; followed by **b** robust postprocessing and quality control steps to identify foci in individual frames, and tracks across multiple (up to 20) frames, which correspond in general to assemblies of labelled molecules. **c** The full extent of photobleaching enables estimates of the characteristic molecular brightness (red arrows), which is narrowly distributed for a fluorescent protein. The characteristic molecular brightness is used to determine **d** total protein number for each region of interest and **e** stoichiometry for each tracked assembly near the start of the image sequence (blue arrows), as a number of molecules (red circles). These metrics are corrected for autofluorescence using unlabelled wild type then collated over a population, enabling robust estimation for average total protein number. **f** Periodicity analysis extracts patterns from the stoichiometry distribution to infer consistent repeat units of assemblies (dark circles). **g** Rapid tracking facilitates analysis of mean-square displacements to estimate individual assembly mobility. **h** Multicolour SlimVar assesses whether stoichiometry and diffusivity are dependent on colocalisation between different pairs of assemblies (white overlap between individual channels in green and magenta).

until an excess of independent single-molecule events has been accrued.

It is possible to estimate stoichiometry by counting photobleaching steps directly[36], but this strategy is typically limited to roughly <10 steps per track due to increasing likelihood of stochastic missing/overlapping steps for higher stoichiometry assemblies. Our ratiometric method instead estimates the stoichiometry of tracks which begin close to the start of the acquisition, typically within a quarter of a photobleaching decay time or less, thus containing minimal prior photobleached content. This approach enables accurate counting over a broader range of stoichiometries than direct step detection.

Independent of the tracking pipeline, SlimVar can quantify the total protein number in the detection volume, or sub-region such as a section of the cell nucleus. We sum all the initial pixel intensities (measured as number of photons) in segmented region of each image and normalise these values by the characteristic brightness corresponding to a single fluorescent dye tag. We define the total protein number as the difference in mean integrated intensity between the test population and that of the unlabelled negative control to corrected for any autofluorescence. We interpret this as the average number of labelled molecules in the volume. It is a reliable population-level measure of protein content and concentration[37].

Since each assembly is associated with a sequence of discontinuous steps along its track, we also quantify the assembly's mean diffusivity using mean-square displacement analysis (Methods). While photobleaching shortens the average track, it transiently improves the optical contrast which is ideal for rapid, high-content tracking. The diffusivity can be used to determine if the assembly is immobile on the timescale of the experiment and therefore likely to be bound, for example to chromatin.

In multicolour SlimVar, multiple channels are captured and tracked independently from the same acquisition. Once the two channels are spatially coaligned, the tracks' spatial and temporal coordinates are compared. This enables measurement of dynamic colocalisation of pairs of tracked particles such as different types of protein that are labelled with different colour dyes. This is a powerful correlative approach which can test the dependence of colocalisation on metrics associated with each assembly, such as stoichiometry and diffusivity.

**Table 1 | Glossary definitions of analysis metrics for single particle tracking**

| Metric/Object | Definition |
|---|---|
| Foci | Spot-like local intensity maxima in a single frame, each corresponding to a localised group of labelled molecules. Associated properties include spatial/temporal location, total intensity, and signal-to-noise ratio. |
| Track | A set of foci in adjacent frames which are spatially close enough to form a contiguous trajectory. Associated properties include those of the set of foci, plus stoichiometry, diffusivity, and signal-to-noise ratio. |
| Characteristic molecular brightness | The average number of photon counts per frame associated with a single fluorescent reporter molecule (e.g. GFP), under a fixed imaging condition. Equivalent to the number of photons in the most common intensity step observed for tracks in the final stage of photobleaching. |
| Integrated nuclear intensity | The total fluorescence intensity, including autofluorescence, of an entire nuclear segment in photon counts, normalised by the characteristic molecular brightness. Dimensionless; described in fluorescent protein equivalents. |
| Total protein number | The average number of labelled molecules in a nucleus, not including autofluorescence, estimated from the difference in integrated nuclear intensity from that of an unlabelled negative control. |
| Stoichiometry | The number of labelled molecules in a track, as estimated by dividing the track's initial intensity by the characteristic single-molecule brightness. |
| Periodicity | The number of labelled molecules in a repeat unit within tracked objects, as estimated by the consistent stoichiometry intervals between nearest-neighbour peaks in the stoichiometry distribution. |
| Diffusivity | An average measure of the rate of random microscopic motion of a track based on the increase in its mean-squared displacement over time. |
| Signal-to-noise ratio (SNR) | A measure of the signal strength of foci or tracks compared to background noise. Higher SNR implies higher confidence of a true positive detection. |
| Sifting | Sifting is the postprocessing step which imposes a minimum SNR threshold on foci and tracks, and a minimum track length, to improve robustness of the track-wise metrics. |

Implementation of single-molecule photobleaching analysis requires the maximum optical contrast, while maintaining both sampling rate and signal-to-noise sufficient to detect foci. The widefield configuration (Fig. 2) maximises fluorescence emission collection from foci within each detection volume. It uses the full back aperture of the objective lens and a minimal number of optical components in the detection path. SlimVar adjusts these elements to mitigate loss of optical contrast at high working depths. These adjustments address either the fluorescence excitation, or the aberrations (blurring of static objects) in the emission path.

The optimal excitation intensity is a few kW/cm² (~10 mW total power within a beam of ~25 μm diameter) which maximises fluorescence emission, whilst being sub-saturating for the dyes and ensuring photobleaching is not too rapid relative to the sampling time. The contrast can be further improved by minimising out-of-focus background. SlimVar avoids exciting the sample outside the detection volume by employing a narrow beam of near-collimated illumination delivered at an oblique-angle similar to variable angle epifluorescence microscopy (VAEM)[38] and highly inclined illumination (HILO)[39,40]. The excitation laser(s) are coaligned and focused into the back focal plane of the high numerical aperture objective lens. To progressively tilt the beam in the sample, this focus is moved to a precise lateral displacement from the optic axis using a steering lens (Fig. 2, point 1) placed conjugate to the back focal plane. The maximum shift of ~3 mm radius would generate a large tilt for TIRF excitation (Fig. 2, inset); for SlimVar, the beam is instead shifted by 2.3 ± 0.2 mm corresponding to a free beam tilted to 60 ± 5° at the sample, subcritical for TIRF. The beam is then stopped down to match the sample dimensions (Fig. 2, point 2) which eliminates further background unrelated to the region of interest.

At low working depths, aberrations in the emission path are corrected by design for most high numerical aperture lenses. However, scattering and aberrations emerge at higher working depths; of these, the most detrimental is spherical aberration caused by differences in refractive index between the immersion medium and the aqueous sample, that compounds rapidly with additional working depth. While in principle SlimVar admits the use of water immersion lenses (NA < 1.3) that minimise this effect, here we adapt the widefield microscope to make more efficient use of a higher numerical aperture oil immersion objective lens with a correction collar (NA = 1.49). These adaptations compensate the optical aberrations at a desired working depth. Using a precision microscope stage, the working depth is set by difference from the axial position where the coverslip surface appears in focus. We present a calibration procedure, inspired by previous works improving optical trapping[41,42] or imaging[43] using high numerical aperture in aqueous samples. It adjusts the optical pathway, including beam collimation, objective correction collar setting and tube lens position, to minimise aberrations at a representative depth of 25 μm (Fig. 2, points 3–4). This calibration improves the optical contrast sufficiently for imaging and tracking across the range of working depths up to 30 μm (Supplementary Figs. 2–5). This procedure can be performed either with a live root sample resting on the coverslip, or a phantom using beads suspended in agarose (Methods). We assessed the improved optical contrast using the width of the point spread functions (PSFs) (Supplementary Fig. 5) and quantified an effective numerical aperture of 1.38 ± 0.02, which though reduced from the diffraction-limited performance, still exceeds that of a water immersion objective lens. Both axial and lateral resolution are improved for beads, and the severe loss of axial performance associated with depth is mitigated in plant roots.

For multicolour experiments, multiple continuous wave lasers are spatially filtered and expanded to the same dimensions, then coaligned using dichroic mirror beamsplitters. Crosstalk and bleedthrough effects tend to decrease the contrast or introduce ambiguous signals; to avoid this we excite and subsequently analyse each channel in alternating interleaved frames (Fig. 2, point 5). This proportionally decreases the effective sampling rate but otherwise maintains the tracking performance of single-colour imaging.

Spatial oversampling, i.e. a small pixel size relative to the widefield spatial resolution, is necessary for postprocessing. It is introduced by additional magnification in the detection path (Fig. 2, point 6). Although each signal is spread more thinly over a greater number of pixels, the lower chance of spurious correlations makes detection more robust to variations in foci shape due to defocus or motion. Oversampling also ensures the localisation precision is not limited by pixel size.

To mitigate detrimental effects of background noise and spatial overlap on the final metrics, sifting is performed to only accept tracks above a minimum set signal-to-noise ratio and track length. Using a dedicated single molecule assay in vitro (Supplementary Fig. 2a, b) and our tracking data from live plants, we found appropriate sifting thresholds determined by two factors: detector noise (Supplementary Fig. 2c) and autofluorescence relative to the probe's characteristic molecular brightness (Supplementary Fig. 6). Sifted tracks have a

## Optical workflow

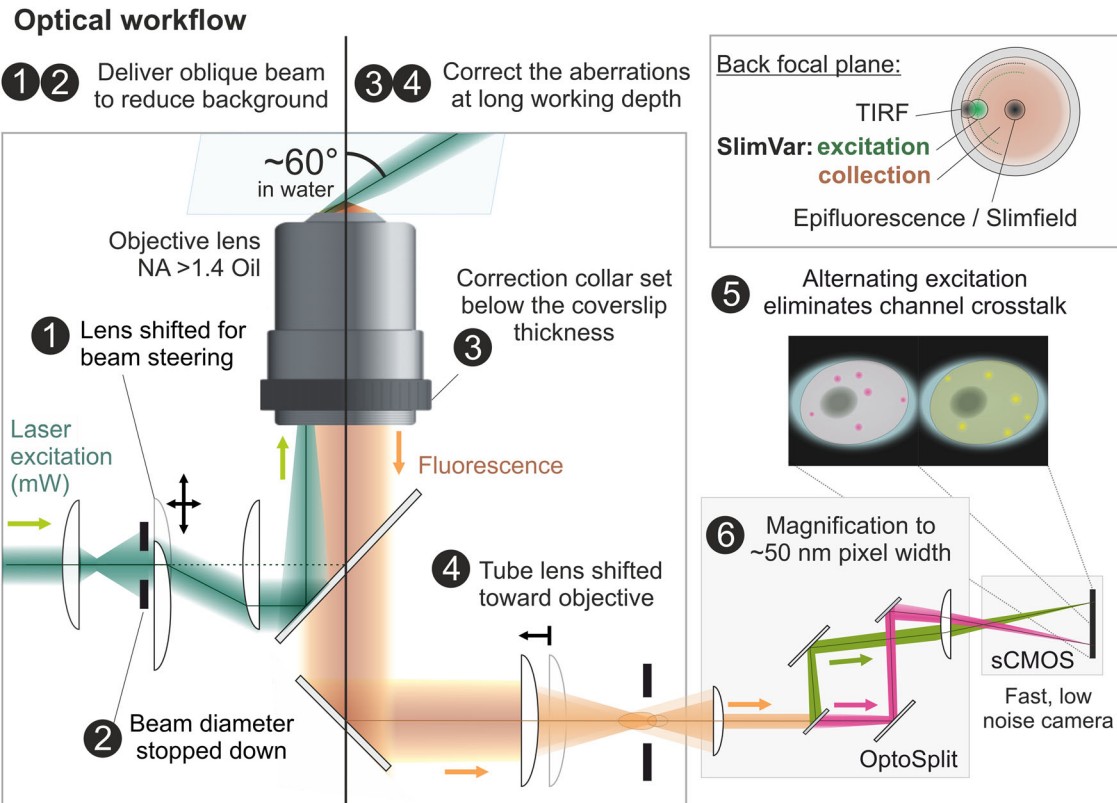

**Fig. 2 | SlimVar enhances optical contrast at greater working depths.** The optical scheme for SlimVar adapts widefield or objective-based TIRF (total internal reflection fluorescence) microscopy capable of detecting single molecules at a coverslip surface, and extends this to greater working depths. A narrow, collimated excitation beam is delivered at a steep but subcritical angle by (1) adjusting the position of a steering lens. The intersection of the focal plane and excitation beam defines a sub-micron high detection volume at the set working depth. The lateral size of this volume can be (2) adjusted using an iris or beam stop to match sample dimensions and reduce background. Aberrations, inherent to oil immersion lenses at depth, are mitigated at the set working depth using a calibration procedure. Either a test sample or an in vitro beads-in-agarose phantom may be used. This comprises a combination of adjustments to (3) an objective lens correction collar and, where necessary, (4) shifting the tube lens towards the objective (Created in BioRender. Payne-Dwyer, A. (2025) https://BioRender.com/13pyw8b). The microscope uses a single detector with a two-colour channel splitter (Cairn OptoSplit); note, the beam is not stopped down after entering the splitter and is shown here with a narrowed diameter only for clarity. In multicolour experiments, contrast is protected from channel crosstalk by (5) alternating excitation wavelengths between subsequent frames. The second pair of lenses in the detection path provides (6) additional magnification (1.2–2.2× depending on physical sensor pixel size) to ensure the point spread function (PSF) is spatially oversampled for super-resolved localisations.

positive predictive value > 95% for single YFPs in vitro (Supplementary Fig. 3a) and >90% for YFP-labelled assemblies in plant roots (Supplementary Fig. 3b).

We assessed the combined improvements to optical and sifting performance using the characteristic molecular brightness of the fluorophore and the total number of photons collected per track. The increase in apparent characteristic molecular brightness with excitation power (Supplementary Fig. 3c), indicates that the detected photon flux associated with single molecule detection at 25 μm depth in vitro is only >50 photons per frame, and therefore >150 photons per single molecule track. Our practical values for characteristic molecular brightness fall in the range 70–200 photons per frame. The total flux emitted is much larger (~1000 photons/ms), with many photons producing the raw image but fewer photons captured in tracks. This is further illustrated by comparing the numbers of photons per track for in vitro control and in vivo data (Supplementary Fig. 4a, b). Modes of 190–400 photons per single molecule track are observed, up to a maximum of ~5000 photons per single molecule. This accounts for ~5% of a typical single fluorescent protein molecule budget of $10^5$ photons[44,45]. This net transmittance corresponds reasonably well to the expected 10%, accumulated from transmittance of the sample (~80% in agarose, ~60% in root tips), effective numerical aperture (29% solid angle), detection optics (~60%) and detector (fill factor >99%, quantum efficiency ~92%). Most tracks have a photon total an order of

magnitude lower, primarily due to the truncating effect of photobleaching (which is accelerated at the high excitation irradiance required), or simply by diffusion out of the detection volume. So, while the observed photon counts in vivo are low compared to theoretical limits of photon budgets for bright fluorescent proteins, they are reasonable considering a realistic budget subject to diffusive, photobleaching and scattering losses.

The fastest diffusivity that SlimVar can detect lies in the range 5–30 μm/s² depending on sampling rate. Only single proteins, if any, exhibit this diffusivity in an intracellular context, and these can still be inferred from the total protein number. Importantly, this suggests that oligomeric assemblies and associated metrics remain representative of the underlying population through sifting, even in cases where the direct detection rate of single molecules is reduced, or where the tracks are truncated. Subject to these limitations, SlimVar is therefore suitable not only for detecting single molecules in plant tissues but also counting molecules within assemblies (stoichiometry and periodicity), and measuring their mobility (diffusivity) and interactions (colocalisation) simultaneously.

### SlimVar resolves dynamic single assemblies of VIN3 and VRN5 in plant nuclei

For examining VIN3 and VRN5 protein localisation and self-assembly behaviours, during and after cold treatment, we utilised lines with

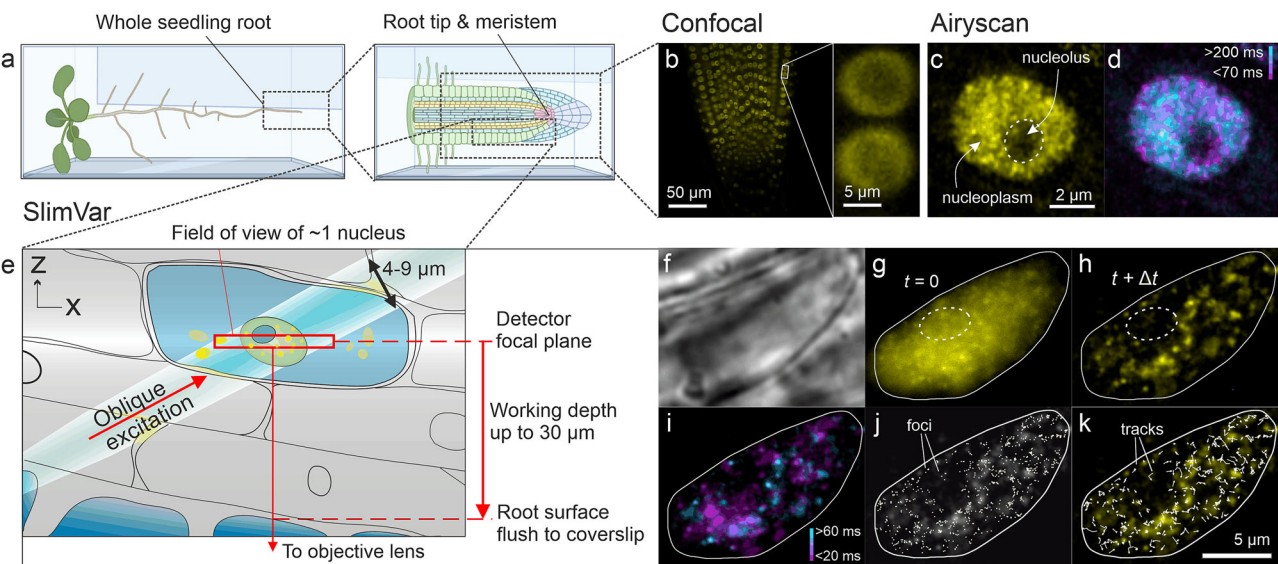

**Fig. 3 | SlimVar resolves dynamics of VIN3 and VRN5 assemblies during cold exposure of root tips. a** Schematic of whole roots laid horizontally in media between agarose and coverslip for confocal and SlimVar microscopy. Created in BioRender. Payne-Dwyer, A. (2025) https://BioRender.com/13pyw8b. **b** Projected confocal z-stacks of VRN5-YFP root tips after 6 weeks of cold; acquisition time 35 s. Insets (interpolated) show VRN5 consistently localised to the nucleoplasm but not the nucleolus. Patterning of VEL proteins appeared round or lens-sh. aped (c.f. Supplementary Fig. 9), with median length 7.8 μm (interquartile range IQR: 5.7–10.3 μm, N = 571), and aspect ratio 1.16 (IQR: 1.06–2.10), comparable to nuclear reporters[78]. **c, d** Airyscan images of VRN5-YFP after 2 weeks' cold indicating heterogeneous distribution, shared scale bar 2 μm; **c** maximum intensity projection of three z-slices, averaged over three consecutive timepoints; **d** residence times estimated from the ratio between median and standard deviation of pixelwise values across three frames. Low standard deviation (cyan) indicates low displacement of foci over 200 ms, equivalent to diffusivity <0.1 μm²/s, while high standard deviation (magenta) indicates high displacement over 70 ms, or diffusivity

>0.3 μm²/s. **e** Schematic indicating illumination and detection volumes (highlighted region and red box, respectively) and working depth. **f–k** SlimVar images of a VRN5-YFP root tip before vernalisation; shared scale bar 5 μm. **f** Brightfield for identifying and centring nuclei; **g** initial fluorescence frame, with nucleolus indicated (white dashes) and overlapping signals; **h** photobleaching transiently increases contrast, revealing distinct assemblies (mean projection of frames 4–6); **i** SlimVar resolves assemblies of different mobility on ms timescales, shown as distinct slow- (cyan, >60 ms residence time, diffusivity <0.4 μm²/s) and fast-moving (magenta, <20 ms residence time, diffusivity >1.4 μm²/s) objects, represented by pixelwise ratio of median and standard deviation of pixelwise values across three frames. **j** Foci are detected from local maxima to super-resolved localisation precision. All sifted foci (Methods) for full sequence shown superimposed (white circles) on panel **h** (greyscale); **k** tracks, generated by linking nearby foci, indicate individual assemblies with independent estimates of stoichiometry and diffusivity. All sifted tracks from sequence shown with one vertex per timepoint (white arrows).

---

VIN3-EGFP/vin3-4 FRI[10,46] or VRN5-EYFP/vrn5-8 JU223[10,16] (referred to as VIN3-GFP and VRN5-YFP respectively) 'with variable transgene copy numbers among progeny due to genetic segregation. To investigate the effects of protein expression and to exploit greater imaging contrast and lower autofluorescence associated with yellow and red fluorophores, we later also characterised new lines containing VIN3-SYFP2/ColFRI (Supplementary Fig. 7) or VRN5-mScarlet-I/vrn5-8 FRI (Supplementary Fig. 8) with different numbers of transgenes. These lines include active FRI alleles so require effective vernalisation for flowering[17].

As a benchmark, we performed traditional confocal microscopy imaging on whole roots (Fig. 3a) with typical sampling at 35 s per frame (Fig. 3b and Supplementary Fig. 9). After identifying nuclei from transmitted light images, we found that both VIN3 and VRN5 exhibited bright but largely diffuse fluorescence localised to the nucleoplasm. We determined the qualitative autofluorescence from the ColFRI negative control; the autofluorescence is greater under 488 nm wavelength excitation but this line shows no signal localisation in either channel (Supplementary Fig. 9). While VRN5 was detectable above this unlabelled background in all nuclei (N = 241) at all timepoints, VIN3 is only discernible during the cold period itself. Its total brightness in nuclei decreased at subsequent timepoints after this, being undetectable within one week after return to warm conditions, as reported[9]. During cold, the VIN3 signal per cell was initially greatest in the vicinity of the meristem and epidermis, before becoming brighter in all cells after further cold exposure (Supplementary Fig. 9).

We then performed Airyscan, an enhanced form of confocal laser scanning microscopy which uses a point detector array; after

optimising the point dwell time, field of view and laser irradiance, we obtained faster frame sampling times down to 60 ms for individual root tip nuclei enabling attempts at video tracking[47]. These image sequences showed a marginally more granular spatial patterning than standard confocal microscopy, hinting at the presence of distinct foci within the diffusive fluorescence (Fig. 3c), with residence times in a similar range to the sampling time (Fig. 3d).

We then implemented SlimVar (Fig. 3e–k) capable of single-molecule fluorescent protein detection within milliseconds, by first identifying nuclei in brightfield (Fig. 3f) to avoid premature photobleaching. We found qualitatively similar nucleoplasmic morphology and localisation (Fig. 3g), but instead of diffuse fluorescence we observed multiple, highly mobile, distinct particles (Fig. 3h) with residence times longer than the exposures, consistent with a sensitivity and sampling speed sufficient to overcome motion blur (Fig. 3i). We thereby detected protein assemblies as distinct foci (Fig. 3j) and connected them into tracks (Fig. 3k).

The excitation beam encapsulates individual cell nuclei between 4–16 μm wide but with minimal excitation of the remaining >70% of the cell volume. It is ideally aligned for the target root tip cells in at least three surface cell layers overlaying the stem cell niche (Fig. 3a and Supplementary Fig. 10), with associated reduction in aberration, backscatter and out-of-focus fluorescence excitation of intermediate cell layers. The contrast available for imaging and tracking indicates that, in principle, cells of different types can be quantitatively discriminated. While SlimVar does not fully recover diffraction-limited resolution in images (Supplementary Fig. 5), the net result is an improvement in our signal-to-noise metric for tracking VRN5

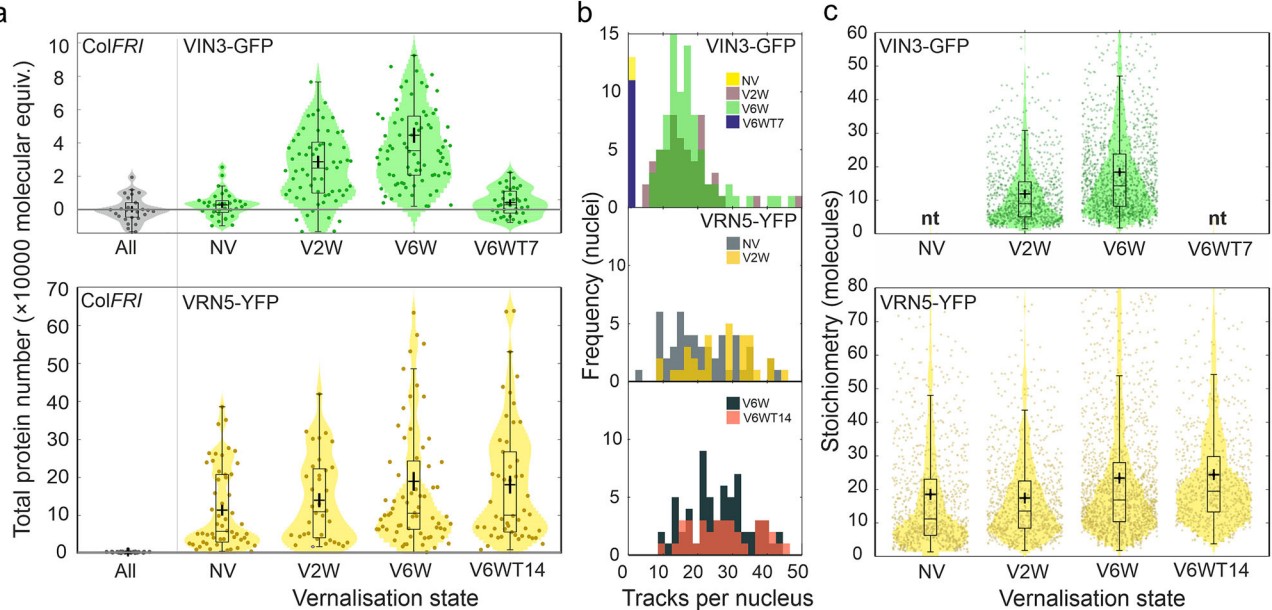

**Fig. 4 | Cold exposure causes VIN3 and VRN5 to form higher stoichiometry assemblies, but only VRN5 assemblies become more numerous. a** Distributions of integrated nuclear intensity (total number of labelled molecules per nucleus prior to correcting for autofluorescence) collated from cells imaged at working depths of $20 \pm 10\,\mu m$ at timepoints before, during and after vernalisation, for VIN3-GFP and VRN5-YFP: NV not vernalised, V2W two weeks of cold, V6W six weeks of cold, V6WT7 six weeks of cold followed by one week of warm conditions, V6WT14 six weeks of cold followed by two weeks of warm. The total protein number is the excess in integrated nuclear intensity above the mean auto-fluorescence in the negative control line, ColFRI (horizontal line). VIN3 total protein number is negligible before vernalisation (two-sided Brunner-Munzel (BM) test vs ColFRI, $N = 33$, $p = 0.11$: not significant at adjusted $p < 0.01$). However, VIN3-GFP increases sharply to ~28,000 ± 3700 molecules after 2 weeks cold ($N = 64$,

$p = 0.0031$), and peaks at ~44,000 ± 4700 after 6 weeks cold ($N = 83$, $p = 6 \times 10^{-7}$). Following transfer to warm conditions, VIN3-GFP reduces to ~3200 ± 1600 molecules within 7 days ($N = 37$, $p = 0.04$). VRN5 levels increase during cold from ~110,000 ± 23,000 to ~190,000 ± 37,000 molecules ($N = 94$, $p = 0.0089$). **b** Numbers of tracks per nucleus (bin width = 2 for clarity; timepoints as in colour legend). VRN5 exhibits an initial increase (NV: 20.8 ± 1.9 up to 26.8 ± 1.6 tracks per nucleus at 2 weeks cold; BM test, $N = 86$, $p = 0.0054$) that is retained (27.0 ± 1.5 and 26.2 ± 2.6 tracks per nucleus at 6 weeks cold and 14 days post-cold respectively; $N = 94$, $p = 0.80$); **c** Collated distributions of stoichiometry (number of labelled molecules per assembly) of individual tracks ($N$ tracks/biological replicates in Supplementary Table 1); nt no tracks detected. Bar, box and whiskers (panels **a**, **c**) denote median, interquartile range (IQR) and ±1.5 IQR respectively; cross: mean ± sem. Source data are provided as a Source Data file.

fluorescent reporters in assemblies by a factor of ~2.6 relative to epi-fluorescence microscopy, including adjustment for the faster exposures (Supplementary Fig. 4c; Brunner-Munzel test, $N = 960$ tracks, $p = 1.1 \times 10^{-5}$, for definition of significance markers see Statistics and Samples). The capability for single-molecule detection is comparable with control samples of purified fluorescent protein (Supplementary Figs. 2–4). The characteristic molecular brightness is sufficiently consistent across the range of working depths to collate acquisitions for each line.

### Vernalisation induces upregulation and self-assembly of VIN3 and VRN5

The visual changes in VIN3 and VRN5 fluorescence during cold exposure suggest corresponding changes in nuclear expression. To test this quantitatively, we acquired SlimVar datasets from nuclei in VRN5-YFP and VIN3-GFP lines. From these, we first determined the characteristic molecular brightness for the GFP and YFP tags (Methods, Supplementary Fig. 1). As a negative control, we imaged nuclei in the ColFRI line using the same 488 nm and 514 nm wavelength excitation modes (Supplementary Fig. 6). We then estimated the total protein numbers in each nucleus of VRN5-YFP and VIN3-GFP (Fig. 4a).

VRN5 was highly abundant at all timepoints, with levels an order of magnitude greater than VIN3 in the VIN3-labelled lines (Fig. 4a). Total VRN5 approximately doubles in response to full vernalisation and persists after return to warm conditions. Total protein numbers translate to nucleoplasmic concentrations of ~100 nM–1 μM for VIN3 and 1–10 μM for VRN5 (Methods). When applied to the cell cytoplasm, the high sensitivity of SlimVar was also able to establish that the fluorescence signals for both VIN3 and VRN5 were marginally above

ColFRI negative control levels, equivalent to a concentration at least 10,000-fold less than those measured in the nucleus.

We also explored the effect of transgene copy number on the abundance of VEL proteins during cold exposure. We generated a homozygous single transgene copy line of VIN3-SYFP2, though not in a deletion background, meaning endogenous VIN3 is also present at a similar level (Supplementary Fig. 7a). The expression of exogenous VIN3 still follows the expected pattern (Supplementary Fig. 9) but at much lower levels than in VIN3-GFP, reflecting the reduction to a single transgene copy of *VIN3*. We used SlimVar to estimate the characteristic molecular brightness for SYFP2 to quantify the total protein number in the VIN3-SYFP2 line (Supplementary Fig. 11). Considering only the labelled VIN3 in the SYFP2 line, the total protein number is 31 ± 4% of that of the VIN3-GFP line at both of the two timepoints. Accounting for the unlabelled copy, this rises to 62 ± 11%. This is consistent with the VIN3-GFP line having three transgene copies, with each copy generating the same amount of protein as the endogenous *VIN3* gene independent of tag.

In both VIN3 fusions, the total protein number exhibits an increase from two to six weeks of cold (Fig. 4a and Supplementary Fig. 11a.), although the distributions partially overlap between the timepoints. The two-week level is between half to two-thirds of the six-week level that approaches full vernalisation. For these lines containing 1–3 copies of VIN3, this steady upregulation does not appear to change from that of a single functional copy.

We then asked whether these cold-dependent increases in protein abundance are evidenced as higher stoichiometry assemblies, or a greater number of assemblies. Identifying these with stoichiometry and number of tracks detected per nucleus respectively, we saw SlimVar as a uniquely powerful tool to address this question in vivo.

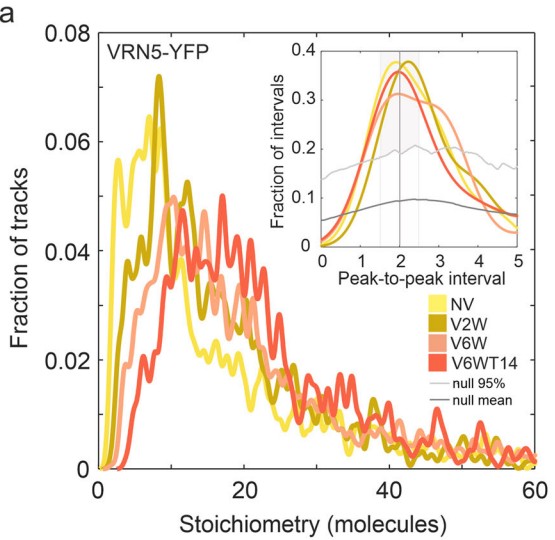
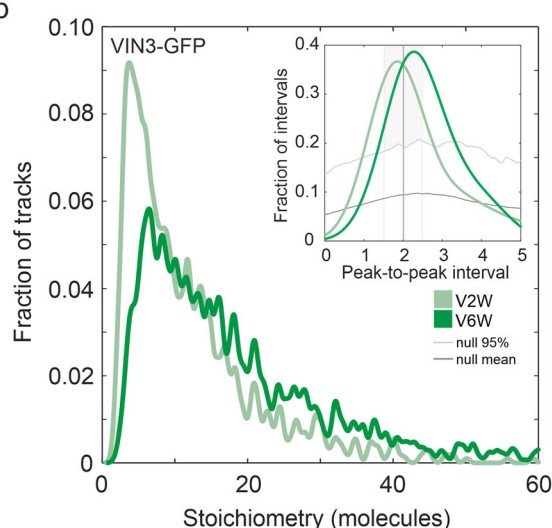

**Fig. 5 | VIN3 and VRN5 assemblies exhibit a two-molecule spacing in their stoichiometry distributions.** The number of labelled molecules in each assembly (stoichiometry) shows consistent peak-to-peak spacing via periodicity analysis of **a** VRN5-YFP and **b** VIN3-GFP across different vernalisation timepoints: NV not vernalised (yellow), V2W two weeks of cold (ochre/light green), V6W six weeks of cold (orange/dark green), V6WT14 six weeks of cold followed by two weeks of warm conditions (red). A kernel width (curve smoothing parameter) of 0.6 molecules was used corresponding to the standard deviation in the observed intensity of a single molecule at the sifting signal-to-noise threshold. Insets: Periodicity analysis - the number of molecules in this subunit can be estimated from the most common spacing between neighbouring peaks in each stoichiometry distribution. The threshold above which a null (aperiodic) distribution can be rejected is the 95th percentile fraction of intervals (grey trace) output from simulated random stoichiometry (Methods). The most common interval is given by the modal kernel density estimate ± s.e.m. above the null threshold (VIN3-GFP: V2W, 1.9 ± 0.3; V6W, 2.2 ± 0.3. VRN5-YFP: NV, 1.9 ± 0.4; V2W, 2.2 ± 0.4; V6W, 2.0 ± 0.3; V6W + T14, 2.0 ± 0.4). The periodic unit in each of these cases is consistent only with an assembly subunit of 2 molecules of either VIN3-GFP or VRN5-YFP. Source data are provided as a Source Data file.

Highly mobile fluorescent foci for both VIN3 and VRN5 could be tracked for up to ~20 consecutive image frames before photobleaching of nuclear contents occurred (Supplementary Movie 1–3). Using bespoke ADEMScode tracking software (Methods) we found that these nucleoplasm-localised protein assemblies were largely excluded from the nucleolus, evident as dark regions 3–6 μm in diameter (Fig. 1k and Supplementary Fig. 8). In larger nuclei, the centres of nucleoli also appeared to exhibit weak VIN3 and VRN5 localisation (Fig. 3b, c and Supplementary Fig. 8). About 1–2% of the total protein number was detected in tracks, which compares well with our estimate, based on the depth-of-field (Supplementary Fig. 2a), that 4–6% of the mean nuclear volume is in sharp focus in each frame.

We detected 10–40 tracks per nucleus in the VIN3 lines during cold exposure, and for VRN5 at all timepoints (Fig. 4b and Supplementary Fig. 11). No tracks were detected either in the Col*FRI* negative control or for VIN3 at pre- or post-vernalised timepoints (Supplementary Table 1). A key finding was that the mean number of VRN5 tracks per nucleus increased by ~30% after the onset of vernalisation and this increase was maintained after return to warm (Fig. 4b). We also considered the number density of these tracks (the number of tracks normalised by the nuclear cross-sectional area). The number density of VRN5 tracks similarly increased by about 30% over the vernalisation time course ($0.55 \pm 0.05$ and $0.71 \pm 0.05$ μm$^{-2}$ before cold and 14 days post-cold respectively: $N = 62$, $p = 0.0042$).

We found that both VIN3 and VRN5 exhibited broad stoichiometry distributions from a few molecules up to several tens of molecules for individual assemblies (Fig. 4c) and that the average stoichiometry increased with time duration spent in the cold during vernalisation. For VIN3, the mean stoichiometry was $12.0 \pm 0.4$ molecules at V2W, increasing to $18.6 \pm 0.5$ molecules at V6W ($N = 1988$, $p = 3 \times 10^{-17}$). For VRN5, assemblies were found to be well developed prior to vernalisation (mean $18.5 \pm 0.6$ molecules at NV). However, there was an increase in stoichiometry during vernalisation which persisted after the return to warm conditions (mean of $24.4 \pm 0.9$ molecules at V6W + T14;

$N = 1626$, $p = 7 \times 10^{-7}$). The greatest change occurred during the intermediate stages of vernalisation between V2W and V6W ($17.4 \pm 0.7$ to $23.4 \pm 0.6$ molecules, $N = 1928$, $p = 4 \times 10^{-14}$).

In summary, the mean number of tracked VRN5 assemblies in each nucleus, and mean stoichiometry of assemblies, each increased over the full course of vernalisation by the same proportion: ~30–35%. Thus, the additional VRN5 protein is divided equally into new vernalisation-induced assemblies, as well as into enlarging assemblies that resembled the pre-vernalised state.

Conversely, VIN3-GFP increased in total protein number by 58% between two-weeks and six-weeks of cold, which matched the proportional 55% increase in its stoichiometry during the same interval. The number of tracks per nucleus did not increase significantly in either of the VIN3 lines (Fig. 4b and Supplementary Fig. 11). All the additional VIN3 protein generated during vernalisation was incorporated into the existing number of assemblies.

## Multimolecular assemblies of VIN3 and VRN5 contain multiples of two molecules

The stoichiometry distributions show a series of periodic peaks (Fig. 5) that are revealed when represented as a kernel density estimate, a method which objectifies the equivalent histogram bin width used[48]. If the assemblies represented have a common oligomeric structure, the characteristic peak-to-peak interval—the periodicity—is equivalent to the number of molecules associated with a physical subunit of the assembly[49]. We developed an analysis method to discriminate this periodicity using the most common nearest-neighbour peak intervals, verified using realistic statistical simulations and experimental data from standard LacI tetramers in vivo[50] (Methods, Supplementary Figs. 12 and 13). Both VIN3 and VRN5 exhibit neighbouring peaks in their stoichiometry distributions separated by two molecules (Fig. 5 insets).

We performed SlimVar on the VIN3-SYFP2 line (Supplementary Fig. 11). Our analysis indicated VIN3-SYFP2 has averages of total protein

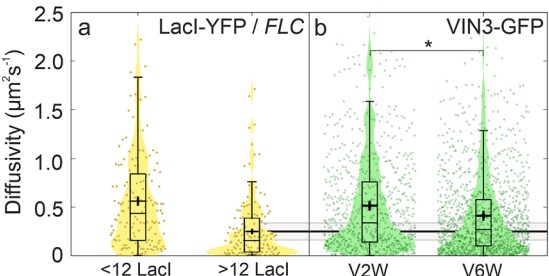
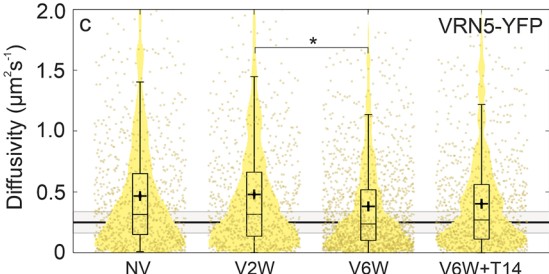

**Fig. 6 | Microscopic diffusivity of VIN3 and VRN5 assemblies decrease towards that of *FLC* loci during vernalisation. a–c** Diffusivity $D$ of individual tracks estimated from mean-square displacement analysis at different vernalisation timepoints: NV not vernalised, V2W two weeks of cold, V6W six weeks of cold, V6WT14 six weeks of cold followed by two weeks of warm conditions. For total numbers of tracks, *N*, see Supplementary Table 1. **a** LacI-YFP tracks of fewer than 12 molecules (*N* = 142 tracks), detected from nuclei without pre-bleaching and identified as unbound LacI, and of LacI-YFP tracks of more than 12 molecules after pre-bleaching, identified as *FLC* candidates (*N* = 153); **b** Diffusivity of VIN3-GFP and

**c** VRN5-YFP before, during and after vernalisation. VIN3 and VRN5 each exhibit a decrease in mobility during the latter part of vernalisation, persisting in VRN5 following return to warm conditions: (VIN3-GFP: $D = 0.52 \pm 0.03$ to $0.41 \pm 0.01\,\mu m^2 s^{-1}$; mean ± sem; $N = 672$ tracks, $p = 0.0011$; VRN5-YFP: $0.47 \pm 0.02$/ $0.48 \pm 0.02\,\mu m^2 s^{-1}$ at NV/V2W to $0.38 \pm 0.01/0.40 \pm 0.02\,\mu m^2 s^{-1}$ at V6W/V6W + T14; $N = 982$, $p = 0.0072$). Horizontal lines denote diffusivity of *FLC-lacO*/LacI-YFP foci under the same conditions (solid line: mean value; grey area, agreement within error). Bar, box and whiskers denote median, interquartile range (IQR) and ±1.5 IQR respectively; cross: mean ± sem. Source data are provided as a Source Data file.

number, stoichiometry and periodicity consistent with the VIN3-GFP line, but only when corrected for the proportion of unlabelled VIN3 present, as estimated from qPCR (Supplementary Fig. 7). We estimated the correction factor by taking the ratio of mRNA expression levels of VIN3 to SYFP2, then normalising by the ratio of VIN3 to GFP expression in the green line, which lacks endogenous VIN3. While inference of quantitative protein levels from mRNA levels is limited, the dependence of the observed periodicity on labelling provides further confidence: if the observed oligomeric species were artefacts mediated by self-interactions of the fluorescent protein tags[51], the periodicity of visible oligomers would not vary in proportion to unlabelled VIN3. Together, these observations are consistent with VEL proteins dimerising within higher-order oligomeric assemblies in vivo. It is intuitive to think of these oligomers growing with the addition of dimeric units, though we cannot test this directly; we do not probe the molecular kinetics or structure that would indicate the pathways of scaffold assembly and disassembly.

**Mobility of larger VIN3 and VRN5 assemblies matches that of *FLC* during cold exposure**

We estimated microscopic diffusivity $D$ for each detected track by calculating the gradient to the initial portion of its corresponding mean square displacement (Methods). For comparison with the diffusivity of *FLC* loci, we performed SlimVar imaging of a *FLC-lacO*/LacI-YFP line with 120 *lacO* copies integrated downstream of the *FLC* transgene[52]. To obtain a qualitative indication of the proportion of VIN3 and VRN5 assemblies which might be bound to *FLC* and their stoichiometry, we analysed just tracks whose diffusivity was comparable to that of a typical *FLC* locus ($D_{FLC} = 0.20\,\mu m^2 s^{-1}$, Supplementary Fig. 8) within individual track measurement error (±0.07 μm²s⁻¹) such that $D$ lies in the range 0.13–0.27 μm²s⁻¹ (Fig. 6a). This simple method of diffusivity matching[53] is particularly helpful when only single-label lines are available. The subset of VIN3 or VRN5 assemblies with diffusivity consistent with *FLC* (Supplementary Table 2) have stoichiometries distributed similarly to the full cohort of VIN3 or VRN5 tracks (Fig. 5c) and show similar increases in stoichiometry over the course of vernalisation for both VIN3 and VRN5. The median stoichiometry of each protein using this diffusivity matching was in the range 10–20 molecules per assembly, increasing with vernalisation.

We found that VIN3 assemblies became significantly less mobile between two and six weeks of cold exposure (Fig. 6b, Supplementary Fig. 11e). VRN5 assemblies exhibited a similar ~20% decrease in diffusivity during the same central stage of vernalisation (Fig. 6c). In keeping with the increase in stoichiometry and lower diffusivity, the proportion of

VIN3 assemblies that show slow *FLC*-like diffusion increases from around 11–15% during early stages of vernalisation up to around 17–18% (Supplementary Table 2). The fraction of *FLC*-like VRN5 assemblies present in the nucleus is already ~18% prior to vernalisation and maintains this level throughout. However, given the greater concentration of VEL protein assemblies compared to *FLC* loci, this proportion is unlikely to be representative of VEL localisations at *FLC*, which if present would be a minority observable only by direct colocalisation.

**VRN5 assemblies at *FLC* have higher stoichiometry during and after vernalisation**

To directly track VRN5 at the *FLC* locus, we generated transgenic plants co-expressing *FLC-lacO* (via LacI-YFP) and VRN5 fused to mScarlet-I (mScI) for dual-colour SlimVar. Our test for colocalisation of VRN5 at *FLC* first required a reliable method to identify *FLC* without perturbing the mScarlet-I reporter for VRN5. For each nucleus, we first performed a rapid *z*-stack using 514 nm wavelength laser excitation to screen for and localise bright, low mobility LacI-YFP foci consistent with *FLC* genomic loci (Fig. 7a). At a chosen *z*-position containing one or more of these *FLC* candidates, we then tracked VRN5-mScI and LacI-YFP using alternating laser excitation (Methods); most images were dominated by the presence of unbound LacI-YFP foci, however, we measured a distinct subset bound to *FLC* (Fig. 7b) with a frequency of $2.3 \pm 1.4$ (mean ± s.d.) per nucleus in the meristem, matching the expectation of 2 *FLC* loci per nucleus within experimental error[52]. This rose to $3.3 \pm 1.7$ in cells toward the transition zone, consistent with more nuclei exhibiting additional pairs of *FLC* loci under genomic endoreduplication. Approximately 40% of detected *FLC* loci were colocalised with VRN5-mScI assemblies, though this proportion was constant across all vernalisation times. This led us to question: might a putative memory element be conditional on the properties of the colocalised VRN5 assemblies, such as stoichiometry?

Both number and stoichiometry of VRN5 assemblies were considerably lower in this single-copy VRN5-mScI line than the multiple-copy VRN5-YFP line (Fig. 4c). Like VRN5-YFP, however, these VRN5 assemblies showed stoichiometries that increased with vernalisation independent of colocalisation at *FLC* (Fig. 7c, grey). There was a proportionally far greater increase in stoichiometry of VRN5 assemblies colocalised at *FLC*, particularly after long cold exposure and after the return to warm (Fig. 7c, magenta). Correspondingly, the fraction of colocalised *FLC* sites associated with assemblies of more than 6 VRN5 molecules increased to ~40% between two and six weeks of cold, representing a ten-fold odds ratio (Fig. 7d). At six weeks' cold, the frequency of *FLC* loci colocalised with >6 VRN5 molecules exceeded an

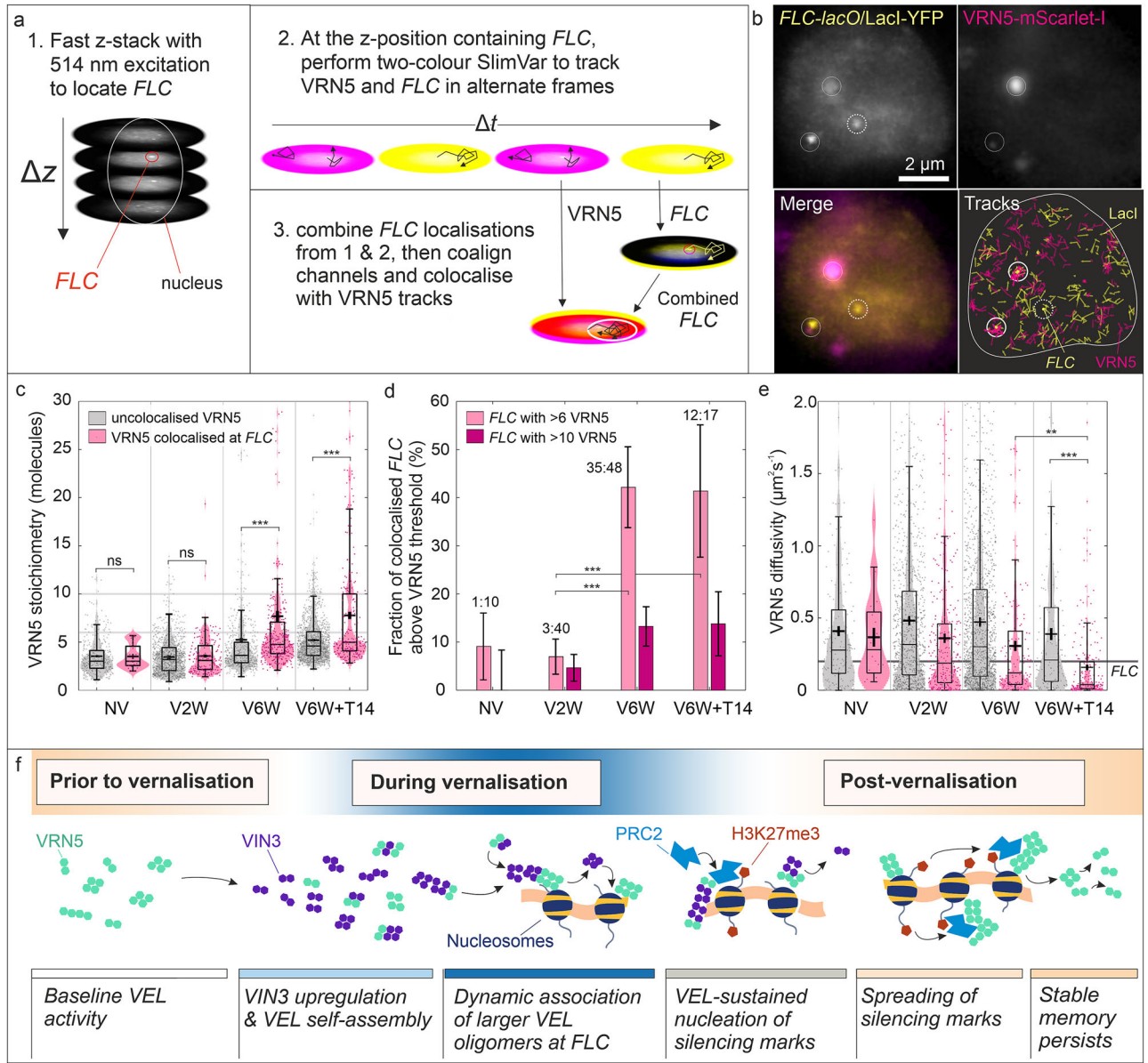

**Fig. 7 | A dynamic subset of enlarged VRN5 assemblies is present at *FLC* loci after long cold exposure and after return to warm. a** Screening for genomic *FLC* loci using a z-stack in the LacI-YFP channel, followed by two-colour alternating excitation in a single *z*-plane to capture **b** foci (dashed circles) and colocalisation events (solid circles) between *FLC* (yellow) and VRN5-mScI assemblies (magenta). **c** The mean stoichiometry of VRN5 when colocalised at *FLC* (magenta) exceeds that of uncolocalised VRN5 (grey) after vernalisation. Timepoints are: NV not vernalised, V2W/V6W two/six weeks of cold, V6W + T14 six weeks of cold followed by two weeks of warm conditions. Bar, box and whiskers denote median, interquartile range (IQR) and ±1.5 IQR respectively; cross: mean ± sem. The difference is negligible before vernalisation ($3.5 \pm 0.3$ vs $3.5 \pm 0.1$, BM test, $N = 365$, $p = 0.39$) but appears at V6W ($7.7 \pm 0.6$ vs $5.0 \pm 0.2$, $N = 2867$, $p = 10^{-18}$) and is sustained for at least two weeks after return to warm ($7.8 \pm 0.5$ vs $5.2 \pm 0.2$, $N = 1416$, $p = 2 \times 10^{-8}$).

**d** Vernalisation preferentially increases the fraction of colocalised *FLC* loci with assemblies of 6 or more VRN5 molecules. Bars denote fractions with square-root estimates of standard error, while exact odds are shown above (for total detected *FLC*, see Supplementary Table 1). A shift to colocalisation with larger VRN5 assemblies occurs between two and six weeks' cold, and remains on return to warm; two-tailed Fisher's exact test: odds ratio OR = 9.7 (2.8–34.0, 95% CI), $p = 7 \times 10^{-4}$ (***) and OR = 9.4 (3.5–25.1), $p = 3 \times 10^{-5}$ (***) respectively. **e** Diffusivity of VRN5-mScI tracks depends on their colocalisation at *FLC*. Only colocalised VRN5 slow to match *FLC* diffusivity during late and post-vernalisation ($0.39 \pm 0.03$ vs $0.16 \pm 0.02$, $N = 309$, $p = 2 \times 10^{-7}$, BM test); Number of tracks, *N*, legend and boxplots as for (**c**). **f** An illustration of the model for VEL-dependent epigenetic memory supported by the imaging results. Source data are provided as a Source Data file.

average of one site per nucleus. This preferential colocalisation of *FLC* with larger VRN5 assemblies after six weeks of cold and on return to warm was detectable for thresholds between 5 and 10 VRN5 molecules.

To establish the timescale of interaction between VRN5 and *FLC*, we then compared their mobility (Fig. 7e). Before cold and at two weeks of cold exposure, the mean diffusivity of the colocalised VRN5 assemblies exceeded that of *FLC* loci by a factor of ~3, indicating short-lived binding relative to the ~10 ms sampling timescale. After six weeks

of cold and after return to warm, the diffusivity of colocalised VRN5 ever more closely matched the typical diffusivity of *FLC* loci, potentially pointing to a tighter interaction. Notably, the fraction of VRN5 not colocalised at *FLC* remained at the higher diffusivity independent of vernalisation. An unexpected observation was the rapid rate of VRN5 turnover at *FLC* that is sensitive to assembly size. The mean apparent residence time at *FLC* of colocalising assemblies smaller than 6 VRN5 was 26 ms before, and 34 ms after, vernalisation, distinctly

shorter than the typical photobleaching time of $63 \pm 2$ ms. For assemblies larger than 6 VRN5 after vernalisation, the residence time was $64 \pm 6$ ms, indicating minimal dissociative loss on this timescale relative to photobleaching. This difference indicates turnover of smaller assemblies at *FLC* on subsecond timescales, in contrast to less transient binding of larger assemblies.

In Lövkvist et al.[14], an assembly size of 17 was predicted as the best fit to observations of nucleating silencing marks, potentially reducing further to 10 molecules in the case of maximal positive cooperativity. Although we do not know whether VIN3 assemblies colocalise specifically with *FLC*, and if so, the size of such an assembly, a large minority of observed VIN3 and VRN5 assemblies with sufficiently low mobility to interact with *FLC* clearly do exceed this size threshold during and after vernalisation. In the case of VRN5, we show this is exceeded by a small minority of assemblies directly colocalised with *FLC* only after vernalisation, though at a single gene copy, the maximal distinction with uncolocalised assemblies occurs at even lower thresholds of as low as 6–10 molecules. This finding suggests that if any lower bound size is required for positive feedback to become active in these colocalised assemblies, it must be very low.

We summarise our findings supported by imaging in an indicative model of epigenetic memory, using *FLC* as the locus (Fig. 7f). It shows VEL protein dynamic self-assembly and function in vernalization-driven epigenetic silencing; prior to vernalisation, VEL proteins interact with each other and PRC2, but the assemblies are modest in size and the majority diffuse too quickly to be dependent on interaction with chromatin. During vernalisation, the VEL assemblies become larger and preferentially localize at *FLC* compared to their status prior to vernalization, consistent with facilitating greater PRC2-mediated nucleation of silencing marks. Large VRN5 assemblies persist at *FLC* after the cold stimulus is removed, which may assist the sustained spreading of marks and maintenance of vernalisation memory.

## Discussion

Here, we have developed an optical microscopy pipeline—SlimVar—and applied it to image cell nuclei in *Arabidopsis* root tips. SlimVar is optimised to detect single fluorescent protein fusions by sampling faster than their typical motion in cells. It therefore enables tracking of molecular assemblies diffusing in comparatively deep, multicellular samples, without requiring complex protocols for chemically conjugating target biomolecules.

In animal tissues, studies using HILO[54] and lattice lightsheet[31] have demonstrated single-molecule tracking to a standard of 50 ms sampling at ~300 μm depth, or 10 ms at ~30 μm depth using dyes, which is comparable to SlimVar with fluorescent proteins. However, in plant tissues, single particle tracking at molecular sensitivity has been demonstrated with TIRF[21,55], or VAEM[56–60] only in the vicinity of a surface cell layer. SlimVar therefore advances the ability to track and count single-molecular assemblies in plants, and potentially in a range of tissues, to that of more complex existing microscopy technologies.

In achieving this speed, SlimVar trades off some of its 3D capability; the detection of foci is restricted to a limited depth of field much smaller than the nucleus. Nonetheless, it is capable of *z*-stacks as used here for systematic *FLC* detection, and it should be possible to extend oblique angle or lightsheet-based approaches to achieve rapid volumetric scans or extended depth of field[61]. The implementation of better index-matching and photon-efficient adaptive optics[62] could also improve the range of accessible working depths currently limited by scattering losses, while also mitigating the required adjustments at each depth. Single-molecule experimental schemes related to SlimVar such as multiple-colour or photoconvertible labelling on the same target[63], may provide promising future avenues to probe the turnover dynamics over longitudinal experiments.

As presented here, SlimVar is implemented on a custom microscope with free-space components. However, the optical components

and controls are essentially similar to commercial widefield/TIRF microscopes used for single-molecule localisation microscopy. Commercial instruments do not typically provide the intense excitation or detector sensitivity sufficient to track single fluorescent proteins at depth (Supplementary Figs. 2–4). As such, we use the platform to explore newly accessible biological insight. In future work, we aim to democratise and enhance SlimVar accessibility further by minimally adapting a representative, modern commercial microscope that meets these requirements. Although optimised for applications in *Arabidopsis* root tips, SlimVar is expected to be adaptable for functional bioimaging research at the molecular scale inside a range of living tissues.

Using SlimVar, we demonstrated oligomeric assemblies of the PRC2 accessory proteins VIN3 and VRN5. VIN3 and VRN5 assemblies increase in stoichiometry above a demonstrable threshold during prolonged cold. Both sets of assemblies exhibit mobility signatures similar to *FLC* gene loci, with a subset of larger VRN5 assemblies clearly demonstrating *FLC* colocalisation. The higher stoichiometry assemblies are therefore prime candidates to contribute to memory element function predicted in a hybrid model coupling protein self-assembly and histone modification[14]. A major advantage of SlimVar over ensemble techniques is that it directly probes not only whether proteins become abundant (total protein number), but whether such changes take effect as higher/lower assembly concentration (tracks per nucleus) or as larger/smaller assemblies (stoichiometry). Our observation that adding more protein subunits preferentially results in a larger average stoichiometry, rather than a greater number of assemblies, is itself consistent with a model of positive cooperativity. Taken together, these findings support the view that VIN3 and VRN5 assemblies mediate epigenetic memory over the extended vernalisation cycle of several weeks (Fig. 7f). However, our current imaging results do not make a direct claim about nucleosomes, only about the *FLC* locus reporter, for which the endogenous locus contains at least three nucleosomes in the nucleation region relevant to VEL function.

The model[14] makes no predictions as to the underpinning factors and mechanisms: neither for protein self-assembly, nor dynamic exchange with the surrounding nucleoplasm. The head-to-tail polymerization via the VEL domain[20] is likely involved, but the predominantly transient interactions between most individual VEL protein assemblies and *FLC* may suggest that the physical feedback processes are more complex than currently understood. Our current interpretation is that the VRN5 assemblies enriched at *FLC*, of sufficient size to satisfy the model, are most likely simple oligomers. Yet, we contemplate whether the very largest of these (~100 VRN5) are instead small, dynamic, phase-separated condensates, related to those observed during transient overexpression of VEL proteins[16,20,64]. If so, the collective, multivalent interactions characteristic of condensates might offer a longer, or otherwise more effective, residence time at *FLC* than we observed for typical molecular assemblies, and therefore a disproportionate contribution to epigenetic memory. SlimVar is an excellent tool capable and primed to further investigate these rare mechanistic events in vivo. Further work will investigate the phenotypic and molecular interactions between VRN5 and VIN3 mutants[64].

While pioneered for the *Arabidopsis FLC* system, these protein-mediated feedback processes may underpin Polycomb-based epigenetic memory common to all eukaryotic systems. Our study demonstrates the interdisciplinary value at the interface of the physical and life sciences of developing SlimVar, and other bioimaging technologies at single molecule precision, to tackle outstanding biological questions including epigenetic processing and memory.

## Methods

### Plant material and growth conditions

VIN3-GFP and VRN5-YFP lines have been described[10,16]. To generate the VIN3-SYFP2 line, *GFP* of the pENTR pVIN3::*VIN3-GFP* construct[65] was

replaced by *SYFP2* by seamless mega-primer cloning. *VIN3-SYFP2* was cloned to the SLJ destination vector (a derivative of SLJ755I5[66]) and transformed to *Agrobacterium tumefaciens* C58 by triparental mating. The transgenic VIN3-SYFP2 plant in Col*FRI* background was generated by floral dipping with *Agrobacterium*. To generate the VRN5-mScarletI line, the *SYFP2* of *VRN5-SYFP2*[64] was replaced by *mScarlet-I* by seamless mega-primer cloning to give pVRN5::*VRN5-mScarlet-I*. This *VRN5-mScarlet-I* was cloned to the SLJ destination vector (a derivative of SLJ6991[66]), transformed to *Agrobacterium tumefaciens* C58 and subsequently transferred into *vrn5-8 FRI* mutants as described above to generate the line. All primers used are listed in Supplementary Table 3. Transgene copy number was determined in T1 or T2 transformants by IDna Genetics (Norwich Research Park). To generate the plant co-expressing VRN5-mScarlet-I and *FLC-lacO*/LacI-YFP, the VRN5-mScarlet-I line was crossed into the *FLC-lacO*/*lacI-YFP* line[52] and was selected by antibiotics.

All seeds were surface-sterilised and sown on 100 mm growth plates containing Murashige and Skoog (MS) medium (Duchefa) with 1 wt.% agar (Difco Bacto) without sucrose. The plates were sealed with Micropore tape (3 M) and kept at 4 °C in the dark for 2–3 days to stratify the seeds. Plates were racked vertically in the growth chamber in warm conditions (16 h light/8 h dark with constant 22 ± 2 °C, 60 ± 10% Relative Humidity (RH)) for 7 days. Non-vernalised time-points (NV) were then imaged on the final day. For all other timepoints, plants were grown in warm conditions for 7 days as above and then were transferred to cold conditions (8 h light/16 h dark, 5 ± 1 °C, 50 ± 30% RH) to vernalise for either 2 or 6 weeks (V2W and V6W respectively). Following vernalisation, a subset of plates was returned to warm conditions for an additional period of either 7 or 14 days (V6W + T7 or V6W + T14). Plates for warm timepoints (NV, V6W + T7, V6W + T14) were handled at room temperature, while those imaged at cold timepoints (V2W and V6W) were transferred on ice to a 4 °C cold room for slide preparation to avoid temperature spikes affecting the fragile vernalised state. VIN3 expression is known to be modulated by the cellular circadian clock[9]; to isolate the long-term trends in expression relating to cold exposure, imaging was performed in day-light hours 4–8 to align with the diurnal maximum in VIN3 expression. At least 3 independent vernalisation courses were grown for each line and timepoint (Supplementary Table 1). Immediately prior to imaging the *FLC-lacO*/LacI-YFP lines (single colour or dually labelled with VRN5-mScarlet-I), LacI-YFP was induced by placing opened growth plates next to a bath of 0.5% ethanol at 25 °C in an airtight container for 2.5 h. This resulted in an optimal amount of LacI-YFP expression in the meristem for SlimVar imaging without spot-bleaching, below the near-saturated induction level (1.5–2% ethanol for 1.5 h[52]) used for confocal study (Supplementary Fig. 12a–d).

### RNA expression analysis

Total RNA was extracted using the phenol method[10,67]. Genomic DNA was removed from the TURBO DNA-free kit (Invitrogen, AM1907) before reverse transcription with SuperScript IV reverse transcriptase (Invitrogen, 18090050) and gene-specific primers. Quantitative PCR analysis was performed on a LightCycler480 II (Roche). Target gene expression was normalised by PP2A (AT1G13320) and UBC (AT5G25760). All primers used in this study are listed in Supplementary Table 3.

### Preparation of samples for imaging

Identical slides were prepared for confocal, Airyscan or SlimVar imaging[47]. Briefly, GeneFrames (Thermo Scientific, AB0578) were fixed to standard slides (VWR) and filled with MS medium plus 1 wt.% agarose to produce agar pads. Where necessary due to seedling size, the terminal >10 mm of the primary root of each plant was excised using a razor. Root tips were laid on each agar pad with tweezers. Liquid MS media was applied to exclude air and each slide was sealed with a plasma-cleaned #1.5 coverslip (VWR). Each slide was imaged within <1 h.

### Confocal and Airyscan imaging

Confocal imaging was performed on a Zeiss LSM880 microscope equipped with argon ion laser and Plan-Apochromat 63×/1.40 NA oil objective lens (Zeiss). Samples were illuminated at 488 or 514 nm wavelengths (GFP or YFP/SYFP2 channels) respectively, and the emission detected at 490–550 nm or 518–550 nm, respectively[47]. The root tip confocal images (Fig. 3b, Supplementary Figs. 9, 12a–d) were acquired as z-stacks over ≤3 z-slices of 1000 × 1000 pixels at 0.6× zoom factor, at 1.5-μm intervals using 20 mW excitation power. Slices were postprocessed in FIJI/ImageJ with a 2D median filter (0.2 μm filter size) to suppress noise before performing a mean z-projection.

Timelapse z-stacks of nuclei (Fig. 3c) were acquired in Airyscan RS mode, after aligning the detector with immobilised TetraSpeck microspheres (Invitrogen, 0.1 μm diameter). Each sequence contained ≤25 volumetric cycles, each taking 0.84 s (12 slices × 56 ms/slice). The axial step size was 0.65 μm with 112 × 112 pixels at 14× zoom factor and 20 mW excitation power. The 3D z-stacks were Airyscan post-processed in Zen Black software (Zeiss) with a user-optimised Wiener deconvolution strength parameter of 5.0. Vernalised timepoints at two and six weeks of cold (V2W, V6W) were imaged at 5 ± 1 °C on a water-recirculating Peltier-cooled stage (Linkam PE100) calibrated using hydrocarbon melting points.

### SlimVar imaging platform

SlimVar was adapted from a Slimfield microscope comprising objective-lens-based total internal reflection fluorescence (Open-Frame, Cairn Research), custom-built from benchtop optical components (Thorlabs) and a nanopositioning stage (Mad City Labs). A general scheme of Slimfield microscopy is available[25,26] with key terms defined in Table 1.

With increasing acquisition depth, refractive index mismatch between immersion oil and aqueous sample is a key challenge[43] due to spherical aberration and excitation beam deviations. This is usually avoided with expensive water or silicone immersion objective lenses, but we demonstrate equal or superior performance with an affordable oil immersion objective lens: NA 1.49 Apo TIRF 100× oil (Nikon). For Slimfield and SlimVar, as with other single-molecule techniques with widefield detection, objective lenses must have a large back aperture diameter, common in lenses specialised for TIRF. Single molecule sensitivity was afforded by a fast sCMOS camera (Teledyne Photometrics Prime95B, 12-bit 'Sensitivity', or BSI, 16-bit 'Sensitivity' i.e. gain of 0.6 photoelectrons per count with an offset of 100 counts). The turret dichroic mirror was either dual-band for GFP/mScarlet-I (Chroma ZT488/561rpc) or for YFP/mScarlet-I (Chroma ZT442/514/561rpc) as appropriate. A 580 nm wavelength longpass beamsplitter (OptoSplit, Cairn Research) after the tube lens ($f$ = 200 mm achromat, ThorLabs AC254-200-A-ML) enabled simultaneous detection of GFP (525/50 nm wavelength centre/bandwidth emission filter) or YFP (550/25 nm) in a green/yellow channel, and mScarlet-I (594/25 nm) in a red channel. Additional magnification (1.2–2.2×) was used to compensate for different camera dexel sizes to maintain an oversampled pixel width of 53 ± 5 nm in the images.

Continuous wave lasers (Coherent OBIS) delivered Gaussian beams ($TEM_{00}$) at 488 nm, 514 nm and 561 nm wavelengths with 1.9 mm FWHM beam diameter that were circularised by an achromatic quarter waveplate[68]. These were steered and focused at the objective back aperture using $f$ = 150 mm lenses to collimate them through the sample with a FWHM diameter of 25 μm.

To optimise background contrast for *Arabidopsis* root tips, the following adaptations were made to the Slimfield microscope:

1. The second convex lens in the expansion telescope was mounted on a lateral translation stage with a high-precision micrometer; this shift (up to 2.3 mm) generated an equivalent lateral

displacement of the beam at the 6 mm diameter objective back aperture, thereby tilting the beam away from the optic axis at the sample. The beam delivery angle was calibrated for lens micrometer position following[36] and set to $50° \pm 3°$ from normal incidence in oil, corresponding to $60° \pm 5°$ in water by Snell's Law, with minimal coupling into the evanescent field at the coverslip, similar to HILO[39] but unlike PaTCH[69] or VAEM[38].

2. A field stop was placed in the conjugate plane upstream of the first telescope lens. Rather than thinning the beam with rectangular slits to further suppress background[40], we chose to maintain a circular beam; this illuminated the full depth of each nucleus for representative estimates of nuclear protein copy and efficient screening of *FLC* loci. The stop was tightened to crop the beam from 25 μm FWHM to between 4 and 9 μm cross-section. The field then approximated an ellipse of $1 \times 3^{1/2}$ the chosen beam cross-section, or 4–16 μm diameter, uniformly illuminated at a power density of 1–5 kW cm$^{-2}$, which is a few-fold less than the saturation excitation intensity of the fluorescent proteins[44].

3. A pair of mirrors were used to incline the beam ~5 mrad away from the optic axis at its intersection with the objective back aperture. This compensates the beam's mismatch-dependent lateral deflection of ~11 μm at the 25 μm calibration depth.

The aberrations were then corrected using the following protocol.

## Optical calibration protocols

Two forms of calibration are presented: basic, relying only on high-signal fluorescence with a correction collar, or an advanced protocol, also using single molecules and tube lens shift.

Basic aberration correction with objective collar only:

1. Choose a nominal working depth (e.g. $d = 25$ μm).
2. Prepare a sample with sub-diffractive fluorescent features at this desired working depth or deeper. This could comprise a suspension of fluorescent beads of diameter <200 nm (Fluospheres, Invitrogen) in 1 wt.% agarose, or weakly autofluorescent point-like features in a root tip.
3. Set the laser excitation to a modest level to reduce photobleaching, e.g. 0.2 mW source power ~0.1 kW/cm$^2$ irradiance and a long exposure time ~100 ms.
4. Move the stage (or objective lens) to focus on the upper surface of the coverslip, then move an additional distance corresponding to the nominal working depth.
5. Move the stage laterally to find a point-like sample feature close to best focus. Centre it in the field of view without refocusing.
6. Acquire a z-stack at <0.2 μm spacing over a maximum range of ±2 μm.
7. Repeat steps 4–6 for several such features.
8. Repeat steps 4–5 to find a new feature. Find best focus using the objective. Iteratively decrease the collar setting to a thinner coverslip setting (by $n_{water}/n_{oil} \times d = 30$ μm, i.e. to ~140 μm) and refocus the objective to achieve best lateral focus (narrowest width of the feature).
9. Repeat steps 4–7 but with the new collar setting, to generate a second set of z-stacks. Compare the differences in PSF (e.g. using MetroloJ QC[70]).

For the advanced protocol, a tilted coverslip sample with YFP (Supplementary Fig. 2a) was created as follows: a $3.2 \times 22$ mm section of #1.5 coverslip was placed between $5 \times 22$ mm sections of adhesive spacer of 260 μm depth (125 μL GeneFrame, ThermoFisher) mounted on a standard $25 \times 75$ mm slide. Additional coverslip sections were added to the spacers to compensate for the thickness of the tilted slip, before sealing with a final coverslip used for imaging. The formed channel was incubated in 1 μg/ml anti-YFP in phosphate buffer saline (PBS) for 5 min, washed with PBS, followed by 50 nM mYFP in PBS for 5 min, then washed with PBS. For imaging, the channel was placed so

that the surface's direction of tilt away from the optical axis was orthogonal to the plane of the beam delivery.

For best results, the advanced protocol requires independent z-positioning of the objective lens and the sample stage, since this enables decoupled correction of both defocus and spherical aberrations.

Advanced aberration correction with single molecule sample and tube lens shift:

1. First, mark the tube lens position. It is best to set any components such as filters between the objective lens and the detector, before −rather than after−performing the calibration.
2. Measure the total image magnification using a graticule.
3. Choose a nominal working depth (e.g. $d = 25$ μm). Prepare samples containing sub-diffractive fluorescent features and slow (or immobilised) single molecules at this desired working depth, such as a dilute (<1 nM) suspension of fluorescent proteins and/or beads in 1 wt.% agarose, or on a tilted coverslip (Supplementary Fig. 2a).
4. Set the laser excitation to a high level, e.g 10 mW power or ~5 kW/cm$^2$ irradiance with 10 ms exposure time. Adjust the camera to the corresponding imaging settings above for the 'balanced mode'.
5. Focus on the coverslip and move to the nominal working depth using the sample stage only (e.g. $d = 25$ μm).
6. Move the stage to a point-like sample feature near focus (this could be a single bead or a section of the volume or surface containing single molecules in focus). Centre it in the field of view.
7. Acquire several fields of view for reference: for beads, acquire z-stacks with <0.2 μm spacing over a range ±2 μm. For fields of view containing visible single molecules, acquire multiple SlimVar sequences with >300 frames; a longer exposure time may be needed to distinguish single-molecule foci from background. Note this exposure time.
8. Iteratively decrease the collar setting to a thinner coverslip setting (by $n_{water} \times d = 30$ μm, i.e. to ~140 μm) and move the stage (not the objective lens if possible) to achieve best lateral focus corresponding to the narrowest width of the feature.
9. Move the objective lens towards the sample (or if this is not possible, move the sample stage towards the objective) a further 20% of the working depth (e.g. by 4 μm to a new total working depth of 29 μm).
10. Compensate by moving the tube lens towards the objective (by up to $\delta = 40$ mm for a 200 mm focal length lens) until the plane of best focus is pulled back onto the chosen feature.
11. Using the stage, re-centre the feature in the field of view. If the region has bleached, pan the stage sideways to a new feature or field of view at the same depth.
12. Then, adjust the collar setting for best lateral resolution as above. Refocus onto the feature (preferably using the objective lens).
13. Repeat the acquisitions in the calibrated state: starting from 10 ms, adjust the exposure time until single molecules are visible. Acquire multiple fields of view in SlimVar sequences containing visible single molecules and track in ADEMScode. Compare the SNR of single molecules. If the exposure time was longer for the reference set, divide the reference SNRs by the square root of the ratio of exposure times. For beads, acquire z-stacks with <0.2 μm spacing. Compare the differences in axial and lateral resolution from the reference state (e.g. using MetroloJ QC).
14. Further iteration of focal position with either collar setting or tube lens position may be desirable for best results, as determined by minimal axial FWHM and maximum signal-to-noise metric.
15. Finally, measure the total image magnification using a graticule, as this may have changed from the nominal design magnification.

We implemented the latter protocol using a combination of in vitro YFP on the tilted coverslip (Supplementary Figs. 2–4) and

fluorescent beads (Supplementary Fig. 5). This introduced a correction collar setting of 140 μm (for a coverslip #1.5H of 170 μm thickness) and a shift of $\delta = -32$ mm in an $f = 200$ mm tube focal length (Supplementary Fig. 5). Despite this shift in the tube lens position, we found the optical pixel size remained consistent within the range $53 \pm 5$ nm, implying a change in magnification no larger than −8%. This is less than the drop of $\delta/f \sim -16\%$ expected from a formal change in tube length, i.e. using a tube lens with a different focal length of $(f + \delta)$ to restore the 4$f$ imaging condition. Since the imaging is improved, the sine condition between the sample and the intermediate image planes must be maintained, which suggests that infinite conjugation breaks down: the emission reaching the camera is no longer parallel to the optical axis at the back aperture of the objective. This also explains the observed reduction in effective numerical aperture by a factor of approximately $(1 - \delta/f)^{1/2} \sim 8\%$ from 1.49 to ~1.38 (Supplementary Fig. 2). We suppose that relaxing the infinite conjugation enables the combined calibration elements to deliver not only the required amount of spherical aberration, but also the correct amount of wavefront defocus[42] for deeper imaging. This result suggests this strategy is workable for small defocus compensations, without introducing a different tube lens or using e.g. spatial light modulators that can provide separable, tuneable corrections to defocus and spherical aberration. However, the beamsplitters and other elements between the objective and tube lens would ideally need to be kept constant between calibrations.

The calibration yields a minimal FWHM of detected foci ~170 nm (Fig. 3d) and a localisation precision[71] of 40–80 nm (Supplementary Fig. 14). In plants we see a net increase of 2.6 in the median signal-to-noise ratio relative to epifluorescence microscopy, when accounting for the decreased exposure time (normalised by the square root of exposure time × excitation power, $N = 500$ foci, Supplementary Fig. 4c).

## Settings for SlimVar imaging in plants

Nuclei were identified in brightfield to find best focus at the nucleolar midbody without photobleaching and were captured for manual segmentation. Fluorescence acquisition settings in green, yellow and/or red channels were pre-optimised to avoid initial camera saturation and to ensure detection of individual tracks of molecular brightness for GFP, SYFP2 / YFP, and mScarlet-I respectively. Three exposure times were used: i) a 'representative' 20 ms providing a low-bias detection of all particles down to single molecules, ii) 'fast' 2 ms to ensure robustness of tracking fidelity and mobility measurements of only the brightest assemblies, and iii) a 'balanced' 10 ms, also capable of single molecule detection. Fast acquisitions captured the subset of assemblies with fourfold (upper quartile) mean stoichiometry at a tripled detection rate due to the faster sampling bandwidth. Results shown derive from data collected with the representative (VRN5-YFP and VIN3-GFP lines) or balanced mode (VIN3-SYFP2, *FLC-lacO*/LacI-YFP and VRN5-mScarlet-I lines) unless otherwise stated. These settings were fixed to minimise systematic variation in the characteristic molecular brightness (Supplementary Fig. 1).

The region of interest spanned ≤300 rows (16 μm) giving a readout time of 2.7 ms per frame, and sampling rates of 44–217 fps. Lasers were triggered in each frame by the camera in 'All Rows' mode to provide global shuttering without extraneous photobleaching. To capture photoblinking, the number of frames in each sequence was set to >50× the photobleaching decay constant.

## Protein tracking, sifting and intensity analysis

SlimVar analysis used batches of OME TIFF image sequences as input to MATLAB-based ADEMScode v2.2 software[72], whose key features are also available in open-source Python package PySTACHIO[73]. The fluorescence sequences were cropped to specify individual nuclei, using masks manually segmented from corresponding brightfield images (code - Segmentation: *mask2seg*). All analysis was restricted to the image region of effectively uniform laser illumination (80% ± 9%

s.d. peak irradiance) no greater than $190 \times 300$ pixels ($10 \times 16$ μm). Each fluorescence sequence was then processed independently (code - Tracking: *trackAllFields*). For each subsequent frame, the local fluorescence maxima-foci-were identified, and their intensity estimated, by integrating the pixel value intensity within 5 pixels, and subtracting a background level averaged over the remainder of a $17 \times 17$-pixel sliding window. Each of the foci was assigned a signal-to-noise ratio (SNR) equal to its intensity divided by the standard deviation of the associated background region. Foci above an initial, permissive SNR threshold of 0.2 were tentatively accepted. These foci were refined to subpixel precision with an elliptical Gaussian masking algorithm. This returned fitted estimates of their semi-axes, which reflected the widefield spatial resolution plus out-of-plane defocus and any residual motion blur. These widefield dimensions (Supplementary Fig. 14) are close to laterally isotropic and insensitive to stoichiometry, since all the assemblies are expected to be much smaller than the diffraction limit and to rotate rapidly relative to the exposure time. The centroids of the foci were also estimated to super-resolved localisation precision[71] of typically 45–80 nm (Supplementary Fig. 14) but as low as 40 nm for bright assemblies, consistent with best Slimfield performance[37]. For each image sequence, pairs of foci were then linked consecutively into tracks if their centroids lay within 8 pixels, with width and intensity ratios in the ranges 0.5–2 and 0.5–3, respectively. When multiple links were possible, the nearest suitable neighbour was chosen.

We then performed sifting of foci and tracks (code - Analysis: *analyseAllTracks*), retaining only those above a strict SNR threshold and minimum number of frames, respectively. The first sifting criterion—an SNR threshold—was robustly determined as the SNR value at which true and false positives occur at equal frequency in a positive control that contains only single molecules; the false positive rate is estimated from negative controls containing only noise. For negative controls, we tracked acquisitions of dark noise (zero excitation intensity), as well as image stacks of simulated noise based on the autofluorescence level measured in wild-type root tips at the typical excitation intensity (Supplementary Fig. 6). Prior works recommend a sifting SNR threshold of 0.4[68,74]; here, we found the appropriate SNR threshold was largely independent of excitation intensity in the kW/cm² range and instead determined by two factors: the autofluorescence relative to the characteristic molecular brightness (Supplementary Figs. 4c and 6), and the detector noise. The actual SNR threshold used ranged from 0.35 for the Prime 95B (12-bit Sensitivity) to 0.50 for the Prime BSI 16-bit mode (Supplementary Fig. 2c).

Even with accurate distinction of foci from background noise, random overlap between tracks increases at high emitter density, leading to spurious summation of their intensities. Noting the temporal correlations present only in real signals, we further strengthened the sifting procedure with a second criterion: we retained only tracks containing at least 3 consecutive foci. This second criterion reduced the true positive detection rate to 35–50% (code – Analysis: *plotTrackFrequency*) but increased the positive predictive value (the mean probability that sifted tracks were correctly identified) from <70% to >95% for single YFP molecules in vitro (Supplementary Fig. 3a). Controls similarly indicate a positive predictive value of >90% for dimers and higher assemblies *in planta* (Supplementary Fig. 3b).

To understand the impact of higher particle densities, we estimated the theoretical probability of random overlap between tracks using a continuum model[74], which estimates the probability of nearest neighbour distances in a random Poisson process falling within the widefield localisation precision. According to this model, the expected fraction of randomly overlapping tracks retained after sifting was <10% for all cases in this study. Conversely, for low densities, the higher the SNR threshold is set, the brighter false positives will appear when they do eventually arise. This can be problematic when the true positive rate

is very low, for example when estimating characteristic molecular brightness at very low emitter density.

The 2D diffusivity of each track was estimated according to a random walk model as a quarter of the rate of increase of the mean-squared displacement with lag time (code – Analysis: *plotDiffusivity*).

The initial intensity of each track was determined by backward extrapolation of the intensities of its first 5 foci to a virgin timepoint prior to photobleaching.

The characteristic molecular brightness of each fluorescent reporter species was determined based on the Chung-Kennedy-filtered terminal intensity of tracks in each acquisition (Supplementary Fig. 1, code – Characteristic Molecular Brightness: *overTrackAll, CKfilterBaseline, plotOverTracks*). After calibration, at 10 ms exposure time in plant nuclei at 20 μm working depth, the characteristic molecular brightness was the following:

VIN3-GFP: $103 \pm 9$ photons ($172 \pm 15$ counts, mean ± sem),
VRN5-YFP and LacI-YFP: $76 \pm 10$ photons ($126 \pm 16$ counts),
VIN3-SYFP2: $79 \pm 15$ photons ($131 \pm 25$ counts) and
VRN5-mScarlet-I: $84 \pm 11$ photons ($140 \pm 18$ counts)

Previous Slimfield work gives comparable values ranging from 60 to 250 photons per frame, albeit for fluorescent proteins within 1 μm from the coverslip surface[35,47,74]. For a quantitative comparison at greater working depths, in vitro values were obtained from recombinant YFP on a tilted surface under the same SlimVar imaging conditions (Supplementary Fig. 4a). The drops in characteristic molecular brightness imply photon collection losses of 25% over the working depth, compared to 30% in the root tissue itself. This corresponds to an average emission decay length in tissue of about 60 μm, which limits the range of working depths across which a given characteristic molecular brightness value can be used within error, to ±10 μm.

The internally calibrated values listed above were then used to normalise intensity metrics into numbers of labelled molecules. First, the number of molecules associated with each tracked assembly – its stoichiometry – is the initial track intensity divided by the characteristic molecular brightness (code – Analysis: *plotStoichiometry*). The same normalisation factor is used for each dataset's stoichiometry periodicity, as well as its total protein number.

## Autofluorescence and total protein number

Raw estimates of the total number of molecules in each nucleus were extracted with an *ImageJ* macro (code – Total Protein Number: *MonoCropper.ijm*); the pixel values were integrated within each nuclear segment, then normalised by the characteristic molecular brightness to give an integrated nuclear intensity in molecular equivalents (Fig. 4a). These values did not account for the additive contribution from autofluorescence background, which we estimated using the corresponding unlabelled control line, Col*FRI*. The total protein number of each labelled dataset was refined to exclude autofluorescence by taking the difference between mean integrated nuclear intensities of the labelled dataset and unlabelled control, adjusted in proportion to the ratio of mean areas of nuclear segments. The negative control was much brighter in green acquisitions ($15,000 \pm 1500$ GFP equivalents, or $1.5 \times 10^6$ photons, vs. $3800 \pm 400$ YFP/SYFP2, or $2.8 \times 10^5$ photons; mean ± sem, $N = 33$, $N = 27$ respectively). The nucleoplasmic concentrations were estimated by dividing each total protein number by the mean nuclear volume (assuming prolate spheroidal nuclei aligned in the image plane) and multiplying by Avogadro's number.

We quantified the mean autofluorescence intensity of nuclei in wild-type control (Col*FRI*) as 45 and 26 photons/pixel for 488 nm and 514 nm wavelength excitation ($N = 250$ and 71 nuclei) respectively. We show that this is not prohibitive for SlimVar in these tissues by considering our SNR metric: if we divide the 103 or 76 photons from a single molecule (characteristic molecular brightness of GFP or YFP respectively) over the average area of 30 pixels per foci, then divide by

the square root of this autofluorescence level, we find expected signal-to-noise ratios of 0.52 and 0.50, which correspond very well to the empirical SNR thresholds 0.35–0.50 used in sifting the single molecules from noise. At the 95th percentile of autofluorescence (109 and 114 photons/pixel), this typical single molecule SNR reduces to 0.24 and 0.33, reflecting the few adverse cases where only oligomers, and not single molecules, can be detected reliably. However, the stoichiometry of those oligomers can still be precisely counted with characteristic molecular brightness obtained elsewhere in regions of sufficiently low local autofluorescence.

## Stoichiometry periodicity analysis

To calculate stoichiometry periodicity (code – Analysis: *plotNearestNeighbourPeriodicity*), first the stoichiometries of all tracks within each nucleus were aggregated across nuclei in a dataset (genotype and vernalisation status), then represented as a kernel density distribution. The choice of the kernel width is informed by the empirical variation in the characteristic molecular brightness. The observed range is typically ±30%, i.e. a width of 0.6 molecules in each frame at the SNR threshold[25], but to avoid oversmoothing during the periodicity analysis, we used the standard error of ±14%, i.e. 0.3 molecules. Peaks in this distribution were detected using the MATLAB *findpeaks* function, and the intervals between nearest neighbour peaks were calculated. The uncertainty in peak-to-peak interval was estimated as the single-molecule uncertainty of 0.6 molecules multiplied by the square root of the mean stoichiometry, divided by the square root of the number of interpolated intervals[69]. To suppress noise and spurious intra-peak sampling, all peak intervals smaller than the interval uncertainty were discarded. A second kernel density estimate was calculated over the remaining peak intervals, with the interval uncertainty as the kernel width. This curve describes the distribution of peak intervals in the stoichiometry as shown in Fig. 5 (insets) and Supplementary Fig. 11d. The modal value of this interval distribution was reported as the predominant periodicity of assemblies in each dataset. This method of estimating periodicity was verified as independent of the mean stoichiometry using simulated positive control data drawn from noisy Poisson-distributed multiples of an oligomeric ground truth[35]. This analysis reproduced the expectation that the minimum number of tracks required for sufficient peak sampling, and therefore the limit of periodicity detection, scales with the square root of the mean stoichiometry. To demonstrate a negative control (code – Analysis: *simulateControlPeriodicity*), 100 aperiodic sets of $10^4$ stoichiometry values, uniformly distributed at random between 1–30 molecules, were generated using the *randperm* MATLAB function and processed to generate a set of 100 independent interval distribution curves each corresponding to null periodicity. The 95th percentile fraction of peak intervals was calculated at each interval size to generate a null curve, below which test data could no longer be considered periodic. The uncertainty in the reported modal peak interval was estimated as the s.e.m. of the peak intervals falling within the range above the null threshold line. To avoid undersampling the peaks, a minimum number of tracks were needed: at least 14 multiplied by the mean stoichiometry. Where necessary, stoichiometry lists from different replicates were aggregated to provide more robust datasets for periodicity analysis. The variation of periodicity for each individual replicate may be estimated by bootstrapping pairs of replicates together to meet the threshold number of tracks. The periodicity analysis was validated in vivo using the standard of tetrameric LacI-YFP[50] detected in the *FLC-lacO*/LacI-YFP lines (Supplementary Fig. 13b).

## Two-colour imaging and colocalisation

For each nucleus, a z-stack was performed, first with brightfield imaging to ensure alignment, then with 514 nm wavelength SlimVar excitation at 10 ms/frame exposure to track *FLC* loci via LacI-YFP. This z-range extended from the highest to lowest surface of each nucleus

with respect to the coverslip surface and was divided into equally spaced intervals of 280–360 nm. During each stack, the $z$-position of the image frame (denoted $I^*$) containing the brightest LacI-YFP foci was noted and subsequently revisited (Fig. 7a). We then performed a dual-colour SlimVar acquisition, 10 ms exposure time alternating between 561 nm and 514 nm excitation wavelengths to facilitate distinct signals for each reporter free from bleed-through. The total duration of the fluorescent $z$-stack and alternating excitation acquisitions for each nucleus was ≤15 s. We estimate that the maximum displacement of $FLC$ loci during the 15 s period was within 180 nm, less than our optical resolution limit (Supplementary Fig. 14d) and consistent with previous observations[75], thus in effect immobile over this timescale.

During post-processing, the two colour channels were spatially aligned to sub-pixel precision using affine transforms generated from SlimVar images of 200 nm diameter fluorescent beads in vitro (code – Alignment: *generateBeadTransform*). Then both yellow and red channel image sequences were tracked independently (Fig. 7b) to generate lists of LacI-YFP and VRN5-mScI tracks respectively. To account for $FLC$ candidates observed in the $z$-stack but photobleached prior to the alternating acquisition, a copy of the alternating sequence was generated in which each yellow channel image simply comprised $I^*$. This copy was also tracked, and its list of LacI-YFP tracks appended to the list from the original sequence. Collected LacI-YFP tracks are shown in Fig. 7b (yellow traces).

To exclude false positive detections of $FLC$ due to free LacI-YFP, we then selected only slow and bright yellow tracks. Our earlier observations suggest a maximum of 8 $FLC$ loci in the entire nucleus (Supplementary Fig. 12). On that basis, we selected the LacI-YFP tracks whose diffusivity was ≤$D_{FLC}$ (Fig. 6a) and from these, retained the 8 brightest (or all if fewer than 8) tracks in each nucleus. These selected $FLC$ tracks, typically 1–4 per nucleus (Fig. 7b, white circles), were run through colocalisation analysis (code – Analysis: *analyseAllTracks*) with the corresponding VRN5-mScI tracks (Fig. 7b, magenta traces) using a reported algorithm[74]. Briefly, VRN5 and $FLC$ tracks were deemed colocalised if they met an intensity overlap condition[76] of at least 50% (effectively a lateral distance of ~3 pixels or one widefield localisation precision) and remained within a distance of 7 pixels (twice the widefield localisation precision) for ≥3 frames. The high numerical aperture and short depth of field ensured an axial precision better than <220 nm FWHM for all colocalisations. The likelihood of false positive overlaps between VRN5 tracks, and the likelihood of false positive colocalisations (an $FLC$ locus being colocalised with a VRN5 assembly by random chance) were both <5%, based on the average initial number density of tracks in each frame (5.2 VRN5 and 3.3 candidate $FLC$) distributed in the nucleoplasm under random point statistics[77]. The residence time of each colocalised track was determined from the number of adjacent colocalised frames (code – Analysis: *plotResidenceTimes*).

## Statistics and reproducibility

Comparisons of the frequency of colocalised assemblies (Fig. 7d) used Fisher's exact test (two-tailed). All other pairwise comparisons used the non-parametric, two-sided Brunner-Munzel test (code – Analysis: *BrunnerMunzelTest*), abbreviated as 'BM test'. Sample size and significance are indicated alongside each result. Investigators were not blinded and each acquisition was independent. We predetermined a target sample size of >24 cells total per line per condition, sufficiently powered to detect changes of <1 s.d. in each of the five test variables (number of tracks, total protein number, stoichiometry, periodicity, diffusivity) at a Bonferroni-adjusted significance level of $\alpha = 0.05/5 = 0.01$. Significance indicators correspond to exact values of $p$ for the relevant test described above, in the ranges $p > 0.01$ (not significant, 'ns'), $0.001 < p < 0.01$ (*), $p < 0.001$ (**) or $p < 0.0001$ (***) respectively. We planned biological replicates of >3 independent cycles of growth and vernalisation, with >3 roots per cycle and >3 cells per root. In the dual line, >10 cells per condition sufficed for estimates of track

number, colocalisation and stoichiometry disaggregated by colocalisation. Technical replicates were identified with tracks detected within each nuclear segment. Actual numbers of replicates analysed are detailed in Supplementary Table 1.

## Reporting summary

Further information on research design is available in the Nature Portfolio Reporting Summary linked to this article.

## Data availability

All raw and processed imaging data and analysed tracks generated in this study are available at the BioImage Archive under accession code S-BIAD1217. Source data are provided with this paper.

## Code availability

ADEMScode v2.2 software for tracking analysis in MATLAB and the corresponding documentation and exemplar data can be found at Github https://github.com/alex-payne-dwyer/single-molecule-tools-alpd with citable version[72] at Zenodo https://doi.org/10.5281/zenodo.16391536.

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

## Acknowledgements

We would like to thank Ibrahim Cissé and Jamie Hobbs for preliminary discussions, Anna Schulten, Govind Menon, Martin Howard, Pan Zhu, Cecilia Lövkvist, Grant Calder and Aisha Syeda for training and discussions, Mathias Nielsen and Yaoxi Li for generating the VRN5-mScarlet-I and VIN3-SYFP2 constructs, and the Horticulture and Bioscience Technology Facility at the University of York for supporting growth chamber resources and confocal microscopy. Funded by EPSRC (grant EP/T00214X/1 to C.D. and grants EP/Y000501/1 and EP/T002166/1 to M.L.). For the purpose of open access, a Creative Commons Attribution (CC BY) licence is applied to any Author Accepted Manuscript version arising from this submission.

## Author contributions

Conceptualization, funding acquisition, administration, supervision; M.L., C.D. Methodology, software and validation: A.P.D., M.L.; Investigation, data curation, analysis and visualization: A.P.D., G.J.J.; Writing – original draft preparation: A.P.D.; Writing – review & editing: A.P.D., G.J.J., C.D., M.L.

## Competing interests

The authors declare no competing interests.
