## [Peer Review file · Nature Communications]

SlimVar for rapid *in vivo* single-molecule tracking of chromatin regulators in plants

Corresponding Author: Professor Mark Leake

Version 0:

Reviewer comments:

Reviewer #1

(Remarks to the Author)

please see attached PDF for review and figures

(Remarks on code availability)

Reviewer #2

(Remarks to the Author)

(Remarks on code availability)

Reviewer #3

(Remarks to the Author)

This manuscript introduces a novel optical microscopy pipeline, SlimVar, used to track two vernalization-specific proteins, VIN3 and VRN5/VIL1, during vernalization. The results observed for VIN3 and VRN5/VIL1 using SlimVar align well with existing knowledge. There are several issues that the authors should address:

1. "Improvement in signal-to-noise for imaging VIN3 and VRN5 fluorescent reporters by a factor of ~3 (Methods)"
The methods section lacks clarity on how the authors calculated the ~3-fold improvement in signal-to-noise. A more detailed explanation would be beneficial.
2. "The number density of VRN5 tracks increased by about a third over the vernalization time course."
There appears to be an increase in transgene expression (RNA levels) during vernalization. The authors should measure the protein levels of VIN3 and VRN5 to determine whether the increase in VRN5 tracks (or the absence of VIN3) is simply due to changes in protein abundance. This is important as the observed intensity of VRN5 (and VIN3) seems to correlate with expression levels.
3. Figure 5f.
The model suggests that VIN3 assemblies, either alone or with VRN5, contribute to oligomer formation at the FLC locus. To gain deeper insights, tracking VRN5 proteins in *vin3* mutants and/or tracking VIN3 proteins in *vrn5* mutants would be highly informative.

(Remarks on code availability)

Reviewer #4

(Remarks to the Author)

The work presented by Payne-Dwyer et al. elucidates the in vivo molecular dynamics of VIN3 and VRN5—two crucial PRC2 accessory proteins involved in FLC silencing during vernalization in Arabidopsis—focusing on their clustering behavior and chromatin association. To achieve this, the authors utilized an enhanced version of their previously published Slimfield microscopy technique, coupled with customized image analysis, to quantify the number of molecules forming VIN3 and VRN5 clusters, track their temporal evolution, assess cluster mobility in response to vernalization, and evaluate their association with the FLC target locus. These findings support a model in which the persistence of large VRN5 clusters facilitates the propagation of H3K27me3 marks even after cold exposure. While the approach and results are highly novel and compelling, the manuscript in its current form is challenging to read and poorly accessible.

I try to summarise the main issues and possible solutions below

(1) The SlimVar imaging modality—a complex approach that is neither as straightforward nor as accessible as the introduction suggests, yet powerful as demonstrated here—, as well as the image analysis approaches are overly concise (expedited) in the main text, lacking the didactic explanations necessary for understanding their principles, purpose, and measurement capabilities. Conversely, the Methods section is dense with technical details that only imaging specialists, well-versed in imaging device engineering, and expert, fluent in programming and knowledgeable in image processing or computer vision, are likely to grasp. While I appreciate the challenge of presenting both the methodological innovation and the associated biological research, the current presentation is very challenging for both microscopists and (plant) cell biologists, relative to the biology and the imaging/image analysis, respectively.

The authors may want to consider dedicating a specific section to thoroughly explain the concept of SlimVar, covering both the imaging modality and image analysis steps, supplemented with relevant illustrations, and diagrams to explain the workflow (steps, concept and purpose; explaining which metrics inform on which molecular aspects; also terms need to be clearly defined at the beginning).

Important for FAIR data sharing: are the design and instructions to reproduce the SlimVar imaging modality available? . Are the scripts and pipelines for image analysis presented at <https://github.com/alex-payne-dwyer/single-molecule-tools-alpd> up-to-date with respect to this work? The authors could refer in the methods to each specific script on Github. For instance I do not see the tool for analysing step-wise photobleaching data, nor periodicity? The time stamps of the last modifications are from 2-6 years ago

(2) The results could benefit from a thorough text revision to make them accessible to both microscopists unfamiliar with vernalization, PRC2 accessory proteins, and chromatin loci, as well as cell biologists not versed in advanced imaging and image analysis. Currently, the types of measurements (e.g., tracks, dwell time, diffusivity-matched stoichiometry, peak spacing of stoichiometry) are often presented abruptly alongside their biological significance (e.g., mobility, turnover, number of molecules) within the same sentence, making the content difficult to grasp on first reading. It would be helpful to break down the rationale more clearly, guiding the reader through the process: first, outline the biological question; next, explain which metrics derived from image analysis are relevant to addressing this question; then, present the numerical results; and finally, discuss the biological significance of these results, breaking down each measurement and figure panel accordingly. Additionally, some figure and table legends lack explanations of the experimental treatments and time points, making terms like V6W, V2W, etc., cryptic for non-specialists. Clearer explanations in these areas would greatly enhance the manuscript's accessibility.

(3) The Model proposes large scaffold of VRN5 at one nucleosome. But can the locus tagged by LacO repeats really resolve one nucleosome? The original paper mentions 120 repeats with 10bp spacing. If one repeat is 30bp (?), this makes a stretch of several kb, hence several nucleosomes? The model and discussion should probably consider this issue.

Below additional comments to consider at revision

- The title hints at a methodological advancement with wide-ranging applications. However, in its current form, the manuscript focuses largely on the biological question. The authors may consider a different title that would fairly convey the main result, while mentioning it was obtained with an innovative imaging approach?

- The authors claim that their approach allows to image deep in the root (epidermis, cortex, endodermis and stem cell niche) with equal resolution. Replicate images, shown as supplements, are required to demonstrate this.

- Some terminology may need better definition or alternatives.

o "protein copy number" does not seem appropriate to me and can be confusing especially when reporting experiments with transgene copy number. "Protein number" shall be sufficient?

o Assemblies should be well defined, currently not very intuitive. I am more familiar with protein cluster, or protein scaffold?

o Stoichiometry is often used alone, but stoichiometry of what? Relative proportion of VIN3/VRN5? Absolute number of VIN3

and/or VRN5 per scaffold? This needs clarification throughout the text.

- Below a list of unclear statements, but this is not an exhaustive list. Perhaps the authors see a pattern that can be considered for clarification throughout the text and figures' legend? A revised version with line number would be helpful.

Abstract

« We report an easy-to-implement method called Variable-angle Slimfield microscopy (SlimVar), which by simple modification of an inverted optical microscope, enables single-molecule tracking of fluorescent reporters in *Arabidopsis thaliana*.» after reading the manuscript, it does not really feel like “a simple modification” only, it is not that much accessible to experimentalists. Also, image processing really requires a strong expertise in programming and image processing. I would thus tone down the simplicity of the whole method. It is powerful yes, but not that affordable to the (plant) cell biology readership.

“ Using SlimVar, we imaged stepwise photobleaching of chromatin-protein assemblies in individual nuclei, 30 μm deep in root tips through multiple cell layers. We find that...»

Adding a short statement connecting photobleaching imaging and single molecule tracking would be useful for the reader's understanding.

“ these assemblies increase in stoichiometry»

this statement is not clear: is it the proportion of VIN3 and VIN5 assembled in a complex that increase in stoichiometry?

Introduction

“ implications for cell differentiation and ageing through to crop security»

I understand the motivation to wrap in a short statement the importance of epigenetic silencing across a broad range of biological studies but this one is a bit abrupt. Would the authors consider expanding a bit on ageing (or consider human health in general, novel therapeutic strategies ?) and about “crop security”? (enhancing crop security and resilience ? agricultural innovations?).

“ A gene where Polycomb Repressive Complex 2 (PRC2) silencing has been well studied is *Arabidopsis* FLOWERING LOCUS C (FLC)” -> “well studied in *Arabidopsis* is FLOWERING LOCUS C ...” ?

“ Cold exposure increases the proportion of FLC alleles that have epigenetically switched from ON to OFF states through nucleation of H3K27me3 at an intragenic site 9,10.» The term “proportion of FLC alleles” can be misleading without further explanation of the tissue scale, since formally speaking there are only two alleles of FLC in the *Arabidopsis* genome. What about “increases the proportion of cells where FLC is epigenetically switched...” ?

“which remains a technical challenge beyond the first cell membrane”. Cell membrane is not an optical challenge, but the cell wall is. ...beyond the first cell wall ? or cell layer?

“Although complex and expensive super-resolution methods including lattice lightsheet 32,33 and MINIFLUX 34,35 are capable of deeper imaging, to date none have been adequately optimised in plants.” I am not sure that “expensive” is a sustainable argument. Also, lattice lightsheet was used in plants (see PMID: 32849711). In addition, I am not sure if “microscopes..optimized “in” plants” is correct. Perhaps “none have been successfully applied for imaging plant tissues in depth, yet”. It is not unlikely that both lattice LSM and MINIFlux are going to be adopted also in plant sciences in the coming years.

Later in the same section: I am not sure the Slimfield imaging modality is commonly available in microscopy facilities, hence does not qualify for a modality at “accessible microscope platforms” as mentioned by the authors. Thus, the use of existing fluorescent reporters and transgenic plant lines might remain the best argument for plant scientists, although it would be fair to explain that a Slimfield modality is also required (not commonly available). The adaptations required for installing this modality (specific lens and lateral beam expansion ?) on a widefield microscope should also be explained together with the basic principles for didactic purposes.

Results

“As an initial benchmark, we used traditional confocal imaging with a typical sampling time of 35 s per frame to examine VIN3 and VRN5 protein localization, during and after cold treatment (Fig. 1b, ColFRI negative control Supplementary Fig. 1).” The results presented in Supp Fig1 should be commented even briefly

“To investigate the effects of protein expression.. we later also characterised new..lines”. How using new protein fusions (different fluorophores) allow to investigate the effects of protein expression? And effects on what kind of measurement/output?

“While VRN5 was detectable well above background levels in all nuclei (N=241) at all timepoints, VIN3 was only discernible at diurnal maxima during the cold period itself.” At that point of reading, the reader does not know what is meant with the “background levels” (technical or biological background?), nor what is the diurnal maxima in this experiment. The reader is not aware that the experiment has been done at specific or different time points during the day? Or does it refer to an experiment, in which case a supplement or a reference is needed.

“During cold, the VIN3 signal per cell was initially greatest in the vicinity of the meristem and epidermis, before becoming

brighter in all cells after further cold exposure.» same here, is there an experiment or a reference to illustrate this statement?
“We then optimised Airyscan» what was optimized? The hardware? The software part? The sample preparation? The acquisition settings?

Figure 1a is cited after Figure 1b. Change the order of the figure panels or cite 1a before in the text. Also, this panel is complex but is described after the conclusion of what it revealed (“.. instead of diffuse fluorescence we observed multiple individual distinct fluorescent foci (Fig. 1d)...”) which makes it difficult to follow. Also, why the foci are “consistent with a greater sensitivity and sampling speed that eradicates motion blur from mobile protein assemblies” is totally obscure. But this issue might be addressed in a restructuring as suggested above.

“The counts per nucleus were normalised by the characteristic intensity due to a single fluorescent protein, determined by single-molecule photobleaching steps in situ “. That photobleaching measurements allow to infer protein counts is not at all intuitive for readers not familiar with the reference 40 and method therein. A short explanation , accompanying also the data presented Supp figure 5, should be provided (ie considering the steps in the loss of intensity – providing enough resolution – to count molecule loss) here or in a different section presenting the entire imaging/image analysis procedure as suggested above.

“we define the protein..copy number ...and identify this with the total number of labelled molecules” – unclear statement , what does “identify” means here compared to “define” ?

“The expression of exogenous VIN3 still follows the expected pattern (Supplementary Fig 1.) but at much lower levels,” lower than what? And later “reflecting the reduction to a single transgene copy of VIN3.” Does this mean that measurements were done in lines with multiple transgene copies? In the same paragraph there is a confusion arising from the use of protein copy number and transgene copy number. Please clarify

“the high sensitivity of SlimVar was also able to establish an upper bound to the fluorescent signal for both VIN3 and VRN5 marginally above the ColFRI negative control levels» what is an “upper bound”? and what is the main information from this cytoplasmic vs nucleus measurement? (VIN3/VRN5 4x more abundant in nucleus?)

Section describing the number of tracks “Using SlimVar, highly motile fluorescent foci for both VIN3 and VRN5 could be tracked ..”. Raw numbers of tracks per nucleus are given. What kind of information does this provide? Wouldn't it be relevant to express these tracks as a percentage of all foci? This would indicate the proportion of mobile “assemblies” /clusters/foci? It would be useful to first ask the question then the method then the result (“to determine whether VIN3/VRN5 are stable or become mobile under xx treatment, we tracked the foci over time...and found...)

“We characterised the average characteristic brightness of a single molecule of the corresponding fluorescent protein (Methods) to determine the bleach-corrected stoichiometry of each track, with which we identified the number of labelled VIN3 or VRN5 molecules in each detected assembly.” This is a complex sentence distracting from the main message coming in the next sentence. Instead it would be useful to explain the question, approach then result (to determine the amount of molecules per cluster/assembly/foci and the changes in composition over treatment we measured...and found...). The same comments are valid for the next sections , hence detailed comments stop here

Figure 4a is cited after 4b. Swap figure panels or respective main text descriptions.

Tables

The legend of each table should explain the abbreviations regarding the treatment / timepoints

Figures

Figure 1 legend

“The nuclear patterning of VIN3 and VRN5 was typically either round or lens-shaped depending on cell type and cell wall attachment”. Has the effect of cell wall attachment on nuclear distribution of VIN3 and VRN5 be demonstrated ? If yes, this statement needs reference. If not remove “and cell wall attachment”

The values presented for panel b indicate several measurements, are they provided in supplemental data? are these measures important for the study? If yes, they should be commented in the main text.

“indicating heterogeneity spatial distribution» -> heterogenous?

Panel d is not so clear. On the one hand, the color scale legend suggests that magenta represent a late time point (>200ms) and cyan an early time point (<70ms) but the legend indicates a color coding according to the mean and the std? why not instead having a legend with only 2 colors: magenta= mean intensity, cyan= deviation from the mean . But this does not show a positive or a negative deviation to the mean – would it be meaningful to consider the 3 categories to feature stable foci over time (over the 3 time points) or foci increasing (assembly?) or decreasing (disassembly?) intensity over time ?

Also, it is not clear why the colored representation of intensities over 3 time points reveal “sensitivity to the dynamics” and especially why plotting the deviation to the mean intensity indicate “motion” (“the latter indicating faster detected motion of VRN5 compared to VIN3 assemblies”). Please clarify / reformulate

Panel g: It is not clear why photobleaching as shown in g reveals distinct assemblies? This image only shows a monochrome (yellow) signal without color coding of intensities over time

Panel h: if this image indeed shows mobility as it is indicated (but the color code is confusing as it is the same as in d which shows differential pixel values=intensities?) then it should be placed after image in panel (i) showing the tracking (as motion and velocity is a result of tracking)

Figure 2

Title “Cold exposure causes VIN3 and VRN5 to form assemblies each of larger, more broadly distributed stoichiometry, and

a relatively modest increase in the number of assemblies". Unclear statement. Please breakdown this combined information: cold causes assemblies, please define "assembly" (cluster of molecules ? accumulation of proteins in a given nuclear focus? Provide the size?); clusters (assemblies) become more variable in size and composition , (explain stoichiometry of what (relative proportion of VIN3/VRN5 in the same assembly? Or amount of protein in one complex assembly?); total number of these clusters moderately increase ?

Legend : why exogenous? The protein fusions are inherently from a non endogenous source.

"The nuclear protein copy number (number of labelled molecules excluding any associated with autofluorescence)" why copy number? Protein number is enough? 2hat means "excluding any associated with autofluorescence"?

Figure 3

Title "VIN3 and VRN5 assemblies exhibit a two-molecule spacing in their stoichiometry distributions." What does this mean?

Does this mean that the scaffold can increase or decrease only (mostly) by two molecules at a time? Please reformulate

"The number of labelled molecules in each assembly shows consistent periodicity of a) VRN5-YFP and b) VIN3-GFP."

Sounds circular to me (a number of molecules showing periodicity of molecules?). please reformulate.

"The fixed kernel width of 0.6 molecules corresponds to the rms error in the observation of a single molecule." What is that?

Where is the value of 0.6 molecules coming from? About what informs the fixed kernel? What is the rms error?

(Remarks on code availability)

I have not reviewed the code functionality as this is not my expertise, but I could see on the Github link provided that the codes seem outdated (last modifications 2-6 years ago), and as mentioned in my comments to the authors, I cannot find all the scripts corresponding to the methods described in the paper - or if they are available, they are not easily findable. The authors should also update the training dataset with one from SlimVar and not only from SlimField as currently on the Github space.

Version 1:

Reviewer comments:

Reviewer #1

(Remarks to the Author)

The authors have satisfactorily addressed all our concerns, and we now recommend the manuscript for publication

(Remarks on code availability)

We have not reviewed this since the revised version, however I dont think this has changed

Reviewer #2

(Remarks to the Author)

(Remarks on code availability)

Reviewer #3

(Remarks to the Author)

I appreciate the thoroughly revised manuscript. I have no further comments.

(Remarks on code availability)

Reviewer #4

(Remarks to the Author)

The authors have invested a lot of work in reshaping the manuscript, including new figures, supp figures and additional explanations. The manuscript remains very dense, but this is due to the nature of the work combining a technological advance applied to answer a biological question, thereby providing a relevant proof-of-concept study.

The new version is very satisfying.

(Remarks on code availability)

Reviewers #1 and #2:

1.1 Optical Description of SlimVar: While the manuscript introduces the new SlimVar microscopy technique, much of the critical optical description is buried in the methods section. Bringing this information forward into the main text would strengthen the manuscript and make it more accessible to a broader readership

Response:

Many thanks to the reviewers – we appreciate the substantive time and effort you have gone to read our manuscript and provide valuable and constructive comments which we have addressed and which improve the clarity of the piece.

We have restructured several sections of the paper, in particular moving the key conceptual and technical descriptions of SlimVar from the Methods into the main body to improve the clarity for a broad range of readers. We now explain the key concepts and limitations at a more comprehensive level as early as possible in the Introduction and initial Results (Figs. 1 and 2, lines 124-351). The remainder of the Methods are ordered sequentially to provide context and clarity for additional technical details and protocols (lines 848-1207).

1.2 Optical Setup Clarity: The optical setup diagram in Fig. 1 is unclear. Specifically, the depiction of the excitation beam being focused parallel to the optical axis but not collinear, alongside the collection of fluorescence across the entire back focal plane (BFP), needs improvement. Please consider redrawing the diagram or providing a more detailed version in the Supplementary Information (SI).

Response:

Thank you for this suggestion. We have split the relatively dense graphical material from the original optical diagram (previous Fig. 1A) into two new figures (Figs. 2 and 3). Fig. 2 is a revised optical diagram outlining the optical hardware elements, properties and tasks that enable the novel SlimVar methods on a widefield microscope. It also explicitly details beam delivery giving more prominence to the near-parallel offset and focusing of the excitation beam, coupled now to a separate text description in Methods (lines 869-960). The back focal plane is now also shown in schematic form indicating the excitation beam diameter and offset and the widefield collection of fluorescence. Fig. 2 visually guides the reader through the concepts, while being supported by clearer technical and numerical details in Methods (including optical component specifications, focal lengths and camera settings). Fig. 3 retains the elements of the original figure specific to plants/VEL proteins and describes the experimental comparison between confocal and SlimVar microscopy.

1.3 Depth Estimation in Imaging: The claim of imaging “30 μm deep into live tissue” in Fig. 1 [*now Fig. 3*] is based on a reference to a previous paper from the senior author, but this remains unclear (“ADEMScode”). More explicit details on how this depth is measured or confirmed is needed.

Response: The indicated working depth up to 30 μm is actually a key new experimental measurement, rather than one cited from earlier literature; apologies, the ADEMScode software citation at this location in the manuscript was a typo which we have now corrected and revised the wording to remove any ambiguity (line 97) and now state (line 258, lines 910 and 934) that the depth is measured by difference from the coverslip with a precision calibrated 3D nanostage (Mad City Labs).

1.4 Refractive Index Mismatch and Imaging Depth: The manuscript does not adequately address the issue of refractive index mismatch at a depth of 30 μm from the coverslip, particularly when using

oil immersion objectives into aqueous media. Typically, such objectives are ineffective at this working distance. The authors should explain how they overcome this challenge and discuss the implications for the effective numerical aperture (NA) and imaging depth.

Response: We strongly value the reviewers' comment – we take your point in that the original submission skipped some of the important technical reasoning of how and why SlimVar is effective. We now explain how SlimVar delivers an improvement over standard implementations using oil or water immersion objectives (lines 250-278). The new Fig. 2 details how we first minimise out-of-focus background fluorescence emission by optimising the angle of tilt and the collimation and diameter of the excitation beam (adjusting the lateral and axial shifts of the focusing lens, and the stop pupil diameter respectively). We then compensate for spherical aberration effects at the chosen depth using a combination of objective lens collar correction, refocusing and where necessary, adjustments to axial position of the tube lens in the detection path (lines 896-996). The cumulative effect of these optimisations enables improvements to optical contrast at higher working depths. When combined with spatial oversampling and robust postprocessing (lines 279-322), this improved contrast enables SlimVar tracking at the precision of single fluorescent protein molecules.

We value the reviewers' suggestion that metrics such as effective numerical aperture (NA) can demonstrate the contrast improvement quantitatively. We now use simple, well-defined performance indicators relative to the ideal diffraction limit at the nominal NA; these are the effective NA and the full-width half-maxima (FWHMs) of the PSF. We have also substantiated several technical details using several new experimental measurements. Using a novel *in vitro* assay employing a tilted microchannel surface (Supplementary Fig. 2) we measure the depth-of-field and from this estimate the effective NA to be approximately 1.38, i.e. superior to a standard high NA water immersion objective lens. We have also directly measured the PSF dimensions from z-stacks of beads suspended in agarose under different calibration and depth conditions (Supplementary Fig. 3). We compare these with dimensions of PSFs extracted from the z-stacks in our existing root tip data (Supplementary Fig. 3) performed as part of the *FLC* colocalisation protocol (Fig. 7a). These new independent quantifications of effective NA and PSF FWHM are consistent with our observation of improved axial resolution and photon collection at depth, beyond the capabilities of a standard high NA water immersion objective lens, or of epifluorescence microscopy with a high NA oil immersion objective lens. We have updated this description both at a clear and simple level at the start of the Results section of the revised manuscript (line 267-272) and at a more granular and technical level in Methods, including an optical calibration protocol (line 896).

1.5 Depth-of-Field Clarification: Current Fig. 1 suggests a depth-of-field of 30 μm , which may be misleading. I think perhaps the authors mean “working distance” here, which is large ($\sim 30 \mu\text{m}$), whereas the focal depth is relatively low with a high NA objective ($\sim 0.5 \mu\text{m}$). Please clarify/define the actual depth-of-field in your experimental setup to avoid confusion.

Response: Yes, we thank the reviewers for spotting this oversight and have changed the label to “working depth” (new Fig. 3) and wording in text here and throughout. We use this specifically to refer to the depth of aqueous sample between the coverslip and the focal plane (rather than the working distance, as the latter includes also the thickness of the coverslip and immersion fluid). At the standard working depth of 25 μm used for SlimVar calibration, we now indicate the experimentally measured depth of field, measured to be approximately 0.53 μm (Supplementary Fig. 2a).

1.6 Tracking Metrics and Photon Quantification: There is limited discussion of tracking metrics, such as the variation in quality with the number of points per track and detected photons. Including a histogram of track lengths would better illustrate the capabilities of SlimVar. Additionally, the manuscript currently reports values in “counts” for photon flux, which should be converted to “photons” to provide a more accurate and interpretable measure.

Response:

We thank the reviewers for this suggestion – we have redrawn all relevant figures (Supplementary Figs. 1, 2 and 4) and statements (e.g. lines 293-313) from ‘counts’ to photons using the camera manufacturer’s gain conversion factor of 0.6 photoelectrons per ADU (i.e. Analog-Digital Units or camera counts). We have revised our description throughout to minimise the use of arbitrary units in favour of detected photons.

We now place our signal-to-noise metric among the early definitions in the manuscript and use it quantitatively in terms of background and photon counts corresponding to single molecule fluorescent tags to justify the quality control thresholds for SNR and minimum track length. These are shown schematically in Fig 1. We now show distributions of track lengths in frames, signal-to-noise, and photons detected per track (Supplementary Fig. 2) and compare these metrics as functions of modality, power and working depth. We include single molecule tracking from our *in vitro* tilted microchannel assay as well as presenting a comparison with the existing datasets from tracking assemblies in plants.

1.7 Autofluorescence Background Quantification: The manuscript lacks quantification of the autofluorescence background, especially deep within the roots, where it is likely to be high at the quoted power densities (kW/cm²). A quantitative characterisation of this background, perhaps using a wild-type control to check for false positives and absolute quantification of autofluorescence photons, would strengthen the findings.

Response: We have added new experimental measurements to quantify the initial autofluorescence background in wild-type *ColFR1* controls based on nuclei segmented in brightfield. We find this lies in the range of a few 10s of photons/pixel with extremes of about 100 photons/pixel (Supplementary Fig. 4). We have also added an analysis of its effect on the single molecule detection threshold (Methods, line 1100-1127). Consistent with this threshold, we observed a nominal rate of false positive foci due to background, but none were accepted into sifted tracks (line 484), the same as for VIN3 lines prior to vernalisation.

1.8 Imaging Data of Single Fluorescent Proteins (FPs): Providing imaging data of single FPs would make the description of SlimVar more robust. Additionally, testing the approach by matching signal-to-background ratio by turning down the laser power until single molecules become difficult to detect would help demonstrate the minimum photon flux detectable by SlimVar. We would expect data showing spatially isolated puncta with single step bleaching (like those shown below) to be readily achieved. The authors mention that “*in vitro* values for characteristic single molecule brightness were obtained from *E. coli* recombinant fluorescent proteins at a coverslip surface under the same SlimVar imaging conditions” but do not provide this data. Including this data, as well as histograms of photon budgets per molecule would emphasise the power of SlimVar and confirm single-molecule detection capabilities.

Response: We thank the reviewers for the valuable suggestion to control for detector noise by sweeping the laser power. Using that approach coupled with a tilted sample geometry (Supplementary Fig. 2a), we now show direct detection of single molecules of purified yellow fluorescent protein YFP at depth (Supplementary Fig. 2b) and, using our metric of characteristic molecular brightness, estimate the required photon flux above background to be >50 photons per 10 ms frame, limited in this case by the detector (Supplementary Fig. 2f). We compare the *in vitro* control to real *in vivo* data (Supplementary Fig. 2e,h). We now discuss this in main text (lines 293-313). To give context for the additional analysis on the yellow fluorescent proteins *in vitro* and in plants, we now cite previous work at lower working depths for a range of fluorescent proteins (line 1085).

1.9 Localisation Precision and Oligomeric Protein Assemblies: The manuscript does not adequately address how the presence of approximately 20 molecules in oligomeric protein assemblies affects localisation precision. It is important to discuss whether the analysis code accounts for deviations from a 2D Gaussian profile in these cases.

Response:

The analysis provides a fit to a 2D unconstrained Gaussian function to generate semi-major/minor axis estimates for all detected foci, with fitted axis lengths typically close to expectations for the widefield optical diffraction limit and relatively insensitive to changes in fluorescent protein stoichiometry; here, the molecular assemblies we detect are much smaller than the diffraction limit and likely rotate ca. 6 orders of magnitude faster than the sampling rate. Differences between the respective two orthogonal Gaussian axes are typically marginal and likely due to small difference in out-of-focal-plane lateral asymmetry in the root tip. We have now added this description in the Methods section (line 1029). We then estimate the super-resolved localisation precision of foci using the standard formulation cited in Thompson et al. (ref. 93), which predicts an approximate scaling of the FWHM of foci with the inverse square root of their photon emission count for relatively small background counts. In practice however, the typical localisation precision we measure (Supplementary Fig. 12) does not reach this limit as most of the localisations occur during later stages of photobleaching and are relatively dim compared to the background counts in the Thompson et al formulation. We add a note to this effect in Supplementary Fig. 12 legend. We note also that improving the lateral resolution of molecular assembly detection has limited practical value without also determining their relative axial positions to similar precision. We have added the explanation that the current implementation of SlimVar does not resolve tracks in the axial direction more finely than the 0.53 μm depth-of-field, though previous Slimfield work indicates SlimVar would be compatible with e.g. astigmatism-based 3D imaging (lines 712-717).

1.10 Single Molecule Imaging in Supplementary Movies: The Supplementary Movies showing single molecules lack contrast, making it difficult to identify them in raw images, especially at long time points. Providing high-contrast grayscale versions of these movies (not in lookup table [LUT] format) would make the signal-to-noise ratio clearer and more convincing.

Response:

The dynamic range of a 16-bit acquisition is naturally challenging to show in 8-bit videos, so any additional visual contrast is valuable. As suggested, we have changed the Supplementary Movies to grayscale.

1.11 Minor Comments: Fig. Clarity: The Fig. panels, particularly 1a (optical setup) and 5a (scanning), are difficult to read and interpret. Improving the clarity and distinguishing features in these panels would make the manuscript more accessible.

Response:

We thank the reviewers for this feedback, and have now split the description of the optical setup across two simpler and clearer figures (Figs. 2 and 3). As we indicate above, the first of these emphasises the experimental geometry and procedures in a much more accessible way than the original submission (See response to comment 1.2). We have also expanded and improved the flow of the figure describing the z-scanning step in the colocalisation experiment (Fig. 7a).

1.12 Colour Usage in Figures: The colours used in Fig. 2, especially in panel b, are hard to distinguish. Using complementary colours instead of different shades would improve accessibility and readability.

Response:

We have now adapted figures for better contrast, including more distinguishable colour schemes in the suggested figure (Fig. 4b).

1.13 Quantification: Please convert all “counts” values to “photons” (i.e SI Fig. 5 and “The characteristic single molecule brightness was 172 ± 15 (counts, mean \pm sem) for VIN3-GFP, 126 ± 16 for VRN5-YFP and LacI-YFP, 131 ± 25 for VIN3-SYFP2 and 140 ± 18 for VRN5” again these values should be measured in photons.

Response: Please see response to comment 1.6.

1.14 Raw Data in Fig. 3: In Fig. 3, it would be helpful to show the raw trace, including the aperiodic data points, in the non-expanded view.

Response:

The insets are not expanded views of the low end of the stoichiometry distribution, but rather the average distribution of intervals between all stoichiometry peaks. To clarify: the ‘raw data’ here for this figure are the tracks’ floating-point stoichiometry estimates (including those not falling into periodic peaks), which are shown in Fig. 4c.

1.15 Dimerisation Wording on Page 8: The wording in the section describing the confirmation that dimerisation was not an artefact of the fluorescent probe (page 8) is confusing. Rewriting this section and elaborating on the plots in SI Fig. 2 would improve clarity for a broader audience.

Response: We have now reworded this description (lines 524-533) to be clearer and, with reference to Supplementary Fig. 5, are more explicit about the underlying calculation and its assumptions.

Reviewer #3 (Remarks to the Author):

This manuscript introduces a novel optical microscopy pipeline, SlimVar, used to track two vernalization-specific proteins, VIN3 and VRN5/VIL1, during vernalization. The results observed for VIN3 and VRN5/VIL1 using SlimVar align well with existing knowledge. There are several issues that the authors should address:

3.1 “*Improvement in signal-to-noise for imaging VIN3 and VRN5 fluorescent reporters by a factor of ~3 (Methods)*”. The methods section lacks clarity on how the authors calculated the ~3-fold improvement in signal-to-noise. A more detailed explanation would be beneficial.

Response:

Many thanks to the reviewer for their constructive comments and suggestions – we have positively responded to all of these to improve the manuscript.

We now state in the Methods how we define and calculate our specific signal-to-noise metric (Table 1 and line 1027 respectively). We have revised this wording in the Results to state the explicit comparison between SlimVar and epifluorescence resulting in a refined estimate of a factor ~2.6-fold for VRN5 in plants (line 400) and now show the data directly (Supplementary Fig. 2i).

3.2 “*The number density of VRN5 tracks increased by about a third over the vernalization time course.*” There appears to be an increase in transgene expression (RNA levels) during vernalization. The authors should measure the protein levels of VIN3 and VRN5 to determine whether the increase in VRN5 tracks (or the absence of VIN3) is simply due to changes in protein abundance. This is important as the observed intensity of VRN5 (and VIN3) seems to correlate with expression levels.

Response:

Assessing the amount of protein directly as a function of vernalisation forms some of the key results in the paper (Fig. 4). Both our confocal (Supplementary Fig. 7) and SlimVar total protein number and tracking results (Fig. 4) corroborate extensive previous biochemical work (lines 62-79) showing cold-dependent increases in expression and protein levels of VIN3, and, to a lesser extent, VRN5. A major advantage of SlimVar over ensemble techniques is that it directly probes not only whether the abundance of tagged protein changes (i.e. the total protein number), but also whether such a change takes effect as higher/lower assembly concentration (i.e. number of detected tracks per nucleus) or as larger/smaller assemblies (i.e. the stoichiometry). We have now added this to the Discussion (line 740).

3.3 Fig. 5f. The model suggests that VIN3 assemblies, either alone or with VRN5, contribute to oligomer formation at the FLC locus. To gain deeper insights, tracking VRN5 proteins in *vin3* mutants and/or tracking VIN3 proteins in *vrn5* mutants would be highly informative.

Response: Our existing experiments are sufficient to demonstrate the potential of SlimVar and the additional memory component at FLC, which is the primary scope of the current manuscript. Nonetheless, we agree that investigating VRN5 in *vin3* / VIN3 in *vrn5* is an intriguing prospect; those lines were planned but did not become available during this research project. However, we highlight in the Discussion (line 760) the value of this suggestion as interesting potential further work. As a further note, mutants of VRN5 and VIN3 are the subject of our separate work in Schulten et al. (now cited as ref. 84 <https://doi.org/10.1101/2024.02.15.580496>) which indicates that VRN5 and VIN3 do heteroligomerise, but have functionally different interaction pathways with PRC2.

Reviewer #4 (Remarks to the Author):

The work presented by Payne-Dwyer et al. elucidates the in vivo molecular dynamics of VIN3 and VRN5—two crucial PRC2 accessory proteins involved in FLC silencing during vernalization in Arabidopsis—focusing on their clustering behavior and chromatin association. To achieve this, the authors utilized an enhanced version of their previously published Slimfield microscopy technique, coupled with customized image analysis, to quantify the number of molecules forming VIN3 and VRN5 clusters, track their temporal evolution, assess cluster mobility in response to vernalization, and evaluate their association with the FLC target locus. These findings support a model in which the persistence of large VRN5 clusters facilitates the propagation of H3K27me3 marks even after cold exposure. While the approach and results are highly novel and compelling, the manuscript in its current form is challenging to read and poorly accessible. I try to summarise the main issues and possible solutions below:

4.1a The SlimVar imaging modality—a complex approach that is neither as straightforward nor as accessible as the introduction suggests, yet powerful as demonstrated here—, as well as the image analysis approaches are overly concise (expedited) in the main text, lacking the didactic explanations necessary for understanding their principles, purpose, and measurement capabilities. Conversely, the Methods section is dense with technical details that only imaging specialists, well-versed in imaging device engineering, and expert, fluent in programming and knowledgeable in image processing or computer vision, are likely to grasp. While I appreciate the challenge of presenting both the methodological innovation and the associated biological research, the current presentation is very challenging for both microscopists and (plant) cell biologists, relative to the biology and the imaging/image analysis, respectively.

Response: Many thanks to the reviewer for their time and effort in appraising our study – we enormously appreciated the comments and opportunities to improve the manuscript’s clarity. We acknowledge the reviewer’s point – at several places in the revised manuscript we have modified the text to improve the simplicity of language and reader accessibility of our SlimVar implementation. Please see also our response to comment 4.2.

4.1b The authors may want to consider dedicating a specific section to thoroughly explain the concept of SlimVar, covering both the imaging modality and image analysis steps, supplemented with relevant illustrations, and diagrams to explain the workflow (steps, concept and purpose; explaining which metrics inform on which molecular aspects; also terms need to be clearly defined at the beginning).

Response:

We have now written a dedicated SlimVar ‘explainer’ section to head up the Results section (line 122) with dedicated workflow figures (Figs. 1 and 2). We now define all key technical terms on their first use as well as in a new glossary table added to Methods (Table 1, line 848).

4.1c Important for FAIR data sharing: are the design and instructions to reproduce the SlimVar imaging modality available?

Response:

Complete instructions to reproduce the modality presented in the explainer section (line 122-292) are now detailed in Methods including the optical train, the calibration protocol and the analysis pipeline (lines 843-1205). In the future we plan to work on an explicit protocol for adapting a commercial widefield microscope in our core technology facility in order to further establish SlimVar as a democratised imaging modality, though we believe this falls into the scope of a separate future publication; we add this note to the Discussion (lines 723-733).

4.1d Are the scripts and pipelines for image analysis presented at <https://github.com/alex-payne-dwyer/single-molecule-tools-aldp> up-to-date with respect to this work? The authors could refer in the methods to each specific script on Github. For instance I do not see the tool for analysing step-wise photobleaching data, nor periodicity? The time stamps of the last modifications are from 2-6 years ago.

Response:

We agree this is an important addition and we thank the reviewer for raising it. We have now revised, updated and documented each script on GitHub and signposted each to its counterpart method in the manuscript as suggested. See also responses to comments R4.46 & 47.

4.2 The results could benefit from a thorough text revision to make them accessible to both microscopists unfamiliar with vernalization, PRC2 accessory proteins, and chromatin loci, as well as cell biologists not versed in advanced imaging and image analysis.

Currently, the types of measurements (e.g., tracks, dwell time, diffusivity-matched stoichiometry, peak spacing of stoichiometry) are often presented abruptly alongside their biological significance (e.g., mobility, turnover, number of molecules) within the same sentence, making the content difficult to grasp on first reading. It would be helpful to break down the rationale more clearly, guiding the reader through the process: first, outline the biological question; next, explain which metrics derived from image analysis are relevant to addressing this question; then, present the numerical results; and finally, discuss the biological significance of these results, breaking down each measurement and Fig. panel accordingly.

Additionally, some Fig. and table legends lack explanations of the experimental treatments and time points, making terms like V6W, V2W, etc., cryptic for non-specialists. Clearer explanations in these areas would greatly enhance the manuscript's accessibility.

Response: These are helpful suggestions, thank you. We have added experimental treatments and time points to each figure and table legend. We have gone through the entire manuscript including figure legends to de-clutter and streamline the text in several places where appropriate. We now also include a glossary in the Methods (Table 1) for all technical terms used.

4.3 The model proposes large scaffold of VRN5 at one nucleosome. But can the locus tagged by LacO repeats really resolve one nucleosome? The original paper mentions 120 repeats with 10bp spacing. If one repeat is 30bp (?), this makes a stretch of several kb, hence several nucleosomes? The model and discussion should probably consider this issue.

Response: We agree that it is interesting to speculate regarding single-nucleosome imaging; the current model (Fig. 7) does not make any direct claim about nucleosomes, only the genetic locus *FLC*, which contains at least 3 nucleosomes in the nucleation region for silencing marks. While the reviewer is correct that the *lacO* repeats (4.7 kb) cannot resolve a single nucleosome (contour length 147 bp, diameter ca. 10 nm), the lateral spatial precision for colocalisation is already fundamentally limited to 70 nm by the lateral spatial precision of the individual foci, so single-nucleosome precision is not something we do or can currently claim to measure. In this regard, we believe this is beyond the scope of our current manuscript. To clarify this, we have edited the experimental description (Fig. 7 legend / line 633) and the discussion (line 758).

4.4 Below additional comments to consider at revision: The title hints at a methodological advancement with wide-ranging applications. However, in its current form, the manuscript focuses largely on the biological question. The authors may consider a different title that would fairly convey the main result, while mentioning it was obtained with an innovative imaging approach?

Response: Thank you for the suggestion. While it is important to indicate the biological result in the title, we believe it is better to prioritise the added value of the SlimVar technique directly, with the multiple changes we have now made to clarify key concepts and technical details of this new technology, including bringing this content forward from the Methods section into the main text as suggested by other reviewers.

4.5 The authors claim that their approach allows to image deep in the root (epidermis, cortex, endodermis and stem cell niche) with equal resolution. Replicate images, shown as supplements, are required to demonstrate this.

Response: We now show example SlimVar images of qualitatively different cell types obtained at different depths (Supplementary Fig. 8). However, we note that in general, cells were not rigorously classified by type using quantitative metrics of cell shape, working depth and distance from the root cap. We have revised the description in the manuscript to better indicate what we specifically mean in this regard: there are qualitative morphological indicators from cells imaged over a range of working depths up to 30 μm including at least three cell layers, suggesting that each of these cell types can in principle be quantitatively discriminated if required (lines 394-399). Importantly, we do not claim that the spatial resolution itself is entirely independent of working depth, as it is corrected at one calibration depth only and is still subject to sample-dependent aberration, leading to differential recovery of the resolution performance in lateral and axial directions (Supplementary Fig. 3). Rather, we show an improvement of the image contrast that is sufficient for single molecule detection up to a maximum working depth of 30 μm , corresponding unambiguously to three cell layers. Although detection is still subject to scattering losses inside the plant tissue, we find that our measurements of stoichiometry

and diffusivity of molecular assemblies are sufficiently consistent to justify collating results across the range of working depths with a characteristic molecular brightness for the fluorescent protein tag. We now reinforce this point in revised wording in the Results (line 399-406).

4.6 Some terminology may need better definition or alternatives. “protein copy number” does not seem appropriate to me and can be confusing especially when reporting experiments with transgene copy number. “Protein number” shall be sufficient? Assemblies should be well defined, currently not very intuitive. I am more familiar with protein cluster, or protein scaffold? Stoichiometry is often used alone, but stoichiometry of what? Relative proportion of VIN3/VRN5? Absolute number of VIN3 and/or VRN5 per scaffold? This needs clarification throughout the text.

Response: We recognise the issues with our original choice in terminology and have chosen to use “total protein number” along the lines of the reviewer’s suggestion. We now define all terms in the new ‘explainer’ section (line 166) and in the glossary (Table 1). Please see also our response to comment 4.41.

4.7 Below a list of unclear statements, but this is not an exhaustive list. Perhaps the authors see a pattern that can be considered for clarification throughout the text and figures' legend? A revised version with line number would be helpful.

Response: Thank you for noting these opportunities to improve the clarity of these statements. We have now added line numbers and have revised the prose and figure legends for clarity.

4.8 Abstract: “*We report an easy-to-implement method called Variable-angle Slimfield microscopy (SlimVar), which by simple modification of an inverted optical microscope, enables single-molecule tracking of fluorescent reporters in Arabidopsis thaliana.*” after reading the manuscript, it does not really feel like “a simple modification” only, it is not that much accessible to experimentalists. Also, image processing really requires a strong expertise in programming and image processing. I would thus tone down the simplicity of the whole method. It is powerful yes, but not that affordable to the (plant) cell biology readership.

Response:

That is an interesting observation of which we are sympathetic, with the following important caveat - SlimVar is implemented on an OpenFrame microscope which does we believe enable levels of accessibility of implementation, though with additional bespoke free-space components beyond this there are indeed more potential technical challenges to overcome requiring some level of intermediate optical microscopy alignment expertise. We have revised the manuscript to better explain the basic principles of SlimVar plus the information required to build a bespoke instrument and to operate it. However, the optical components and controls are essentially similar to commercial widefield/TIRF microscopes used for single molecule localisation. Historically, these commercial setups have not always provided the intense excitation or detector sensitivity sufficient to track single fluorescent proteins at depth, to the requirements that we now demonstrate in Supplementary Fig. 2. As such, the main remit of this work was to use the bespoke platform to explore the newly accessible biological insight, rather than to broaden the implementation. However, the gap in commercial performance has significantly narrowed; in future work, we aim to construct and share a more accessible version of SlimVar by minimally adapting a representative, modern commercial microscope, e.g. Zeiss Elyra 7, that meets these requirements, which we now note in the Discussion (line 726-736).

Reflecting on the reviewer’s feedback, we have also toned down our original language as to SlimVar’s ease of use (lines 15-18, 702-736, 847-861). We also value the feedback that the use of existing

fluorescent reporters and transgenic plant lines remains one of the best arguments for plant scientists to adopt SlimVar (lines 93-106). Please also see our response to comment 4.16.

4.9 *“Using SlimVar, we imaged stepwise photobleaching of chromatin-protein assemblies in individual nuclei, 30 μm deep in root tips through multiple cell layers. We find that...”*

Adding a short statement connecting photobleaching imaging and single molecule tracking would be useful for the reader's understanding.

Response: We have revised the Abstract (line 15), Introduction (line 103) and Results (lines 148, 176) to include the explicit link between controlled photobleaching and tracking with molecular steps in intensity.

4.10 *“these assemblies increase in stoichiometry”* - this statement is not clear: is it the proportion of VIN3 and VIN5 assembled in a complex that increase in stoichiometry?

Response:

We agree with the reviewer – the term ‘stoichiometry’ used in this original context was ambiguous. To explicitly clarify, we have changed the wording to “Upon cold exposure, the number of assembly molecules increases up to 100% to a median of ~20 molecules.”

4.11 Introduction. *“implications for cell differentiation and ageing through to crop security”* - I understand the motivation to wrap in a short statement the importance of epigenetic silencing across a broad range of biological studies but this one is a bit abrupt. Would the authors consider expanding a bit on ageing (or consider human health in general, novel therapeutic strategies ?) and about “crop security”? (enhancing crop security and resilience ? agricultural innovations?).

Response: Thank you - we have now expanded the opening statements to explore the suggested topics concisely (lines 30-37).

4.12 *“A gene where Polycomb Repressive Complex 2 (PRC2) silencing has been well studied is Arabidopsis FLOWERING LOCUS C (FLC)”* -> “well studied in Arabidopsis is FLOWERING LOCUS C ...” ?

Response: We have now reworded this as suggested (line 42).

4.13 *“Cold exposure increases the proportion of FLC alleles that have epigenetically switched from ON to OFF states through nucleation of H3K27me3 at an intragenic site 9,10.”* The term “proportion of FLC alleles” can be misleading without further explanation of the tissue scale, since formally speaking there are only two alleles of FLC in the Arabidopsis genome. What about “increases the proportion of cells where FLC is epigenetically switched...” ?

Response: As the *FLC* loci switch independently even in the same cell, we have rephrased (line 45) to “Cold exposure increases the probability that each FLC locus will epigenetically switch from ON to OFF states through nucleation of H3K27me3 at an intragenic site.”

4.14 *“which remains a technical challenge beyond the first cell membrane”*. Cell membrane is not an optical challenge, but the cell wall is. ...beyond the first cell wall ? or cell layer?

Response: We have changed this text to “which remains a technical challenge beyond the first cell layer” (line 86).

4.15 “Although complex and expensive super-resolution methods including lattice lightsheet 32,33 and MINFLUX 34,35 are capable of deeper imaging, to date none have been adequately optimised in plants.” I am not sure that “expensive” is a sustainable argument. Also, lattice lightsheet was used in plants (see PMID: 32849711). In addition, I am not sure if “microscopes..optimized “in” plants” is correct. Perhaps “none have been successfully applied for imaging plant tissues in depth, yet”. It is not unlikely that both lattice LSM and MINFLUX are going to be adopted also in plant sciences in the coming years.

Response: We have changed this to “Although complex, expensive super-resolution methods including lattice lightsheet and MINFLUX are in principle capable of deeper imaging, to date neither has been successfully applied at a molecular scale in plants.” (line 87).

4.16 Later in the same section: I am not sure the Slimfield imaging modality is commonly available in microscopy facilities, hence does not qualify for a modality at “accessible microscope platforms” as mentioned by the authors. Thus, the use of existing fluorescent reporters and transgenic plant lines might remain the best argument for plant scientists, although it would be fair to explain that a Slimfield modality is also required (not commonly available). The adaptations required for installing this modality (specific lens and lateral beam expansion?) on a widefield microscope should also be explained together with the basic principles for didactic purposes.

Response: Please see our response to comment 4.8.

4.17 Results. “As an initial benchmark, we used traditional confocal imaging with a typical sampling time of 35 s per frame to examine VIN3 and VRN5 protein localization, during and after cold treatment (Fig. 1b, ColFRI negative control Supplementary Fig. 1).” The results presented in Supp Fig1 should be commented even briefly

Response: We have moved this text down to where the ColFRI background is mentioned (line 333-339), and we have elaborated on the important result (Supplementary Fig. 7). Please also see also our response to comments 4.20 and 4.34.

4.18 “To investigate the effects of protein expression.. we later also characterised new lines”. How using new protein fusions (different fluorophores) allow to investigate the effects of protein expression? And effects on what kind of measurement/output?

Response:

These lines are phenotypically functional but contain different transgene copy numbers (lines 332, 452). This section aimed to establish whether a different copy number of VIN3, or differently labelled fusions of VIN3, affect the expression and total protein number. Our finding is that the three forms of VIN3 (endogenous, GFP fusion and SYFP2 fusions) all have similar levels of expression per copy and upregulation response to cold exposure. Since the wild type has one copy, SYFP2 has two and GFP we reasonably infer to have three (lines 452-465), VIN3 function and regulation do not appear to be finely dependent on the precise concentration above a single functional copy. We have modified the text at this point to make this finding clearer (line 467).

4.19 “While VRN5 was detectable well above background levels in all nuclei (N=241) at all timepoints, VIN3 was only discernible at diurnal maxima during the cold period itself.” At that point of reading, the reader does not know what is meant with the “background levels” (technical or biological background?), nor what is the diurnal maxima in this experiment. The reader is not aware that the experiment has been done at specific or different time points during the day? Or does it refer to an experiment, in which case a supplement or a reference is needed.

Response:

We have added explanation that the "background level" is the mean intensity due to cellular autofluorescence (as quantified in Supplementary Fig. 4). We note that VIN3 has an auxiliary role in the winter-time circadian clock, which does require the daily timepoints of experiments to be fixed. This is stated in the Methods (lines 803-805), as it is an important factor for reproducibility, but since it is neither varied nor helpful to the key manuscript narrative, we have migrated its mention away from the main text.

4.20 "During cold, the VIN3 signal per cell was initially greatest in the vicinity of the meristem and epidermis, before becoming brighter in all cells after further cold exposure." same here, is there an experiment or a reference to illustrate this statement?

Response:

The VIN3 panels in our confocal microscopy imaging illustrate this change (Supplementary Fig. 7). We have now added a reference to this figure (line 344).

4.21 "We then optimised Airyscan» what was optimized? The hardware? The software part? The sample preparation? The acquisition settings?

Response:

Specifically, we optimised the acquisition settings (ROI, laser power, scan rate, postprocessing strength) for the highest possible frame rate at the required signal intensity. We have now edited the text to reflect this and separated the motivation for using Airyscan from the subsequent optimisation (line 345-34).

4.22 Fig. 1a is cited after Fig. 1b. Change the order of the Fig. panels or cite 1a before in the text. Also, this panel is complex but is described after the conclusion of what it revealed (".. *instead of diffuse fluorescence we observed multiple individual distinct fluorescent foci (Fig. 1d)...*") which makes it difficult to follow.

Response:

We have now thoroughly reorganised the figure panels and the references to them in this section (lines 325-392), and have ensured that all these panels get cited sequentially in the main text.

4.23 Also, why the foci are "*consistent with a greater sensitivity and sampling speed that eradicates motion blur from mobile protein assemblies*" is totally obscure. But this issue might be addressed in a restructuring as suggested above.

Response:

We have changed this to "we observed multiple, highly mobile, distinct particles with residence times longer than the exposures, consistent with a sensitivity and sampling speed sufficient to overcome motion blur. We thereby detected protein assemblies as distinct foci and connected them into tracks." (line 389). For better clarity we now describe the need to overcome motion blur in the new 'explainer' section of the manuscript (line 127).

4.24 "*The counts per nucleus were normalised by the characteristic intensity due to a single fluorescent protein, determined by single-molecule photobleaching steps in situ*". That photobleaching measurements allow to infer protein counts is not at all intuitive for readers not familiar with the reference 40 and method therein. A short explanation, accompanying also the data presented Supp Fig. 5, should be provided (i.e. considering the steps in the loss of intensity – providing enough resolution – to count molecule loss) here or in a different section presenting the entire imaging/image analysis procedure as suggested above.

Response:

We thank the reviewer for highlighting the need articulating for the link between photobleaching and counting and that this needs to be made more explicit. We now address this in the new 'explainer' section at the start of the Results (Fig. 1) and have added a short explanation of stepwise photobleaching to the relevant figure (now Supplementary Fig. 1) as suggested.

4.25 *"we define the protein copy number ...and identify this with the total number of labelled molecules"* – unclear statement , what does "identify" means here compared to "define" ?

Response:

We have now moved this text to the 'explainer' section (lines 170-173) and changed the wording to 'interpret' rather than 'identify', so it is clear which is the definition and which is the usage.

4.26 *"The expression of exogenous VIN3 still follows the expected pattern (Supplementary Fig. 1.) but at much lower levels,"* lower than what?

Response:

We have corrected this to state that the level of expression is lower than in the line with multiple copies of GFP-labelled VIN3 (line 455).

4.27 And later "reflecting the reduction to a single transgene copy of VIN3." Does this mean that measurements were done in lines with multiple transgene copies? In the same paragraph there is a confusion arising from the use of protein copy number and transgene copy number. Please clarify

Response:

Yes, that is correct - these measurements are done with different copies. We now clarify what the gene copy and protein number are for each specific line (line 453-463).

4.28 *"the high sensitivity of SlimVar was also able to establish an upper bound to the fluorescent signal for both VIN3 and VRN5 marginally above the ColFRI negative control levels"* what is an "upper bound"? and what is the main information from this cytoplasmic vs nucleus measurement? (VIN3/VRN5 4x more abundant in nucleus?)

Response:

We have now rephrased this text (line 420) to the less ambiguous sentence: "When applied to the cell cytoplasm, the high sensitivity of SlimVar was also able to establish that the fluorescence signals for both VIN3 and VRN5 were marginally above ColFRI negative control levels, equivalent to a concentration at least 10,000-fold less than those measured in the nucleus."

4.29 Section describing the number of tracks: *"Using SlimVar, highly motile fluorescent foci for both VIN3 and VRN5 could be tracked .."*. Raw numbers of tracks per nucleus are given. What kind of information does this provide? Wouldn't it be relevant to express these tracks as a percentage of all foci? This would indicate the proportion of mobile "assemblies" /clusters/foci? It would be useful to first ask the question then the method then the result ("to determine whether VIN3/VRN5 are stable or become mobile under xx treatment, we tracked the foci over time...and found...)

Response:

We thank the reviewer for pointing out that the results section for track number and stoichiometry needed clearer motivation in the text. We have rewritten this (line 469) to start with the question of "whether these cold-dependent increases in protein abundance are evidenced as higher stoichiometry assemblies, or a greater number of assemblies." We note that the tracks correspond to a very small

minority of foci in a cell and are subject to a low detection rate, with an implication that it is more helpful to work in numbers per nucleus rather than risk dividing by low numbers. We later also show the proportions of mobile and immobile tracks using *FLC* mobility as a reference. (Fig. 6 and Supplementary Table 1).

4.30 “*We characterised the average characteristic brightness of a single molecule of the corresponding fluorescent protein (Methods) to determine the bleach-corrected stoichiometry of each track, with which we identified the number of labelled VIN3 or VRN5 molecules in each detected assembly.*” This is a complex sentence distracting from the main message coming in the next sentence. Instead it would be useful to explain the question, approach then result (to determine the amount of molecules per cluster/assembly/foci and the changes in composition over treatment we measured...and found...). The same comments are valid for the next sections, hence detailed comments stop here

Response:

We have removed this sentence as it is no longer needed, thanks to the definitions being moved earlier, and have also simplified the text in the sections that follow.

4.31 Fig. 4a is cited after 4b. Swap Fig. panels or respective main text descriptions.

Response: We have now swapped the order in which these are discussed/cited (lines 465, 484).

4.32 Tables: The legend of each table should explain the abbreviations regarding the treatment / timepoints

Response: Please see our response to comment 4.2.

4.33 Figures Fig. 1 legend: “*The nuclear patterning of VIN3 and VRN5 was typically either round or lens-shaped depending on cell type and cell wall attachment*”. Has the effect of cell wall attachment on nuclear distribution of VIN3 and VRN5 be demonstrated? If yes, this statement needs reference. If not remove “*and cell wall attachment*”

Response:

We have removed the text 'cell wall attachment' as suggested (line 360), as that relationship has not been systematically investigated.

4.34 The values presented for panel b indicate several measurements, are they provided in supplemental data? are these measures important for the study? If yes, they should be commented in the main text.

Response: We cite the confocal data (Supplementary Fig. 7) more thoroughly and discuss these in the Results (line 333-345) as they contain important further examples of imaging that establish the negative control and the trend in expression. We have now added a cross reference to Fig. 3b. The values stated in the Fig. 3b legend are indicative of the ranges of morphological indicators for cells that were later imaged and collated in SlimVar (now stated in line 389), though these descriptors are not directly investigated.

Please see also our responses to comments 4.5, 4.17 and 4.20.

4.35 “*indicating heterogeneity spatial distribution*» -> heterogenous?

Response: We have now corrected this typo (line 365).

4.36 Panel d is not so clear. On the one hand, the color scale legend suggests that magenta represent a late time point (>200ms) and cyan an early time point (<70ms) but the legend indicates a color coding according to the mean and the std? why not instead having a legend with only 2 colors: magenta= mean intensity, cyan= deviation from the mean . But this does not show a positive or a negative deviation to the mean – would it be meaningful to consider the 3 categories to feature stable foci over time (over the 3 time points) or foci increasing (assembly?) or decreasing (disassembly?) intensity over time ?

Response:

We have now edited the figure caption (now Fig. 3) to state more clearly that the ratio between the median and standard deviation of the signal in each pixel over several frames corresponds to the residence time (ms) of the assemblies generating that signal. Since local intensity maxima have a fixed characteristic width resolved by the microscope, these residence times can also be expressed as diffusivities ($\mu\text{m}^2/\text{s}$). Thus, the ratio offers a metric for how rapidly particles diffuse, and the sensitivity of each imaging technique to particle motion, without objectively identifying each particle. We chose to represent this pixel-wise estimate as a residence time, not as a diffusivity, in the figure annotations, since it is important for it not to be confused with track-based estimates of diffusivity which are reported later in the manuscript. However, we now indicate the corresponding diffusivities in the figure caption. We did not show the signed deviations over time because these are confounded by the direction of motion (drop in signal behind a particle, increase in signal in front), rather than indicating assembly or disassembly of the particles.

4.37 Also, it is not clear why the colored representation of intensities over 3 time points reveal “sensitivity to the dynamics” and especially why plotting the deviation to the mean intensity indicate “motion” (“the latter indicating faster detected motion of VRN5 compared to VIN3 assemblies”). Please clarify / reformulate

Response: Please see our response to comment 4.36.

4.38 Panel g: It is not clear why photobleaching as shown in g reveals distinct assemblies? This image only shows a monochrome (yellow) signal without color coding of intensities over time

Response:

We now clarify in the new ‘explainer’ section (line 176) and the Fig. 3 caption (line 375) that the initial stages of photobleaching improve contrast rapidly by removing some background, resulting in the superior contrast in panel Fig. 3h compared to panel Fig. 3g.

4.39 Panel h: if this image indeed shows mobility as it is indicated (but the color code is confusing as it is the same as in d which shows differential pixel values=intensities?) then it should be placed after image in panel (i) showing the tracking (as motion and velocity is a result of tracking)

Response:

The pixels in the image (now panel 3k in the revised manuscript) show the average intensity over 3 frames, not velocity or mobility. They are the same pixels as shown in the current panel 3h-i. We have rewritten the caption (line 381) to state that the overlaid tracks (which themselves indicate mobility across a longer timespan) come from the full sequence of image frames.

4.40 Fig. 2. Title “Cold exposure causes VIN3 and VRN5 to form assemblies each of larger, more broadly distributed stoichiometry, and a relatively modest increase in the number of assemblies”.

Unclear statement. Please breakdown this combined information: cold causes assemblies, please define “assembly” (cluster of molecules ? accumulation of proteins in a given nuclear focus? Provide the size?); clusters (assemblies) become more variable in size and composition , (explain

stoichiometry of what (relative proportion of VIN3/VRN5 in the same assembly? Or amount of protein in one complex assembly?); total number of these clusters moderately increase ?

Response:

We have now revised this figure title to “Cold exposure causes VIN3 and VRN5 to form higher stoichiometry assemblies, but only VRN5 assemblies increase in number”. We now define the term ‘assemblies’ as early as possible in the manuscript (line 130): each assembly is an accumulation of labelled proteins in a given spatiotemporal focus, but also inferred from their co-tracking over time to have a high probability of being continually clustered together. We now also define our use of ‘stoichiometry’ as the ratiometric measurement of the number of molecules in an assembly (line 140).

4.41 Legend : why exogenous? The protein fusions are inherently from a non endogenous source.

Response: We have now removed the term ‘exogenous’ (line 439).

4.42 “The nuclear protein copy number (number of labelled molecules excluding any associated with autofluorescence)” why copy number? Protein number is enough? That means “excluding any associated with autofluorescence”?

Response:

We have now changed this terminology throughout the manuscript to ‘total protein number’ to avoid confusion with transgene copy number and defined it at the start of the Results section (line 166 and Fig. 1). We have rephrased the figure legend to read “The total protein number is calculated as the excess of the integrated nuclear intensity above the mean level of the autofluorescence background in the negative control line” (line 436).

4.43 Fig. 3. Title “VIN3 and VRN5 assemblies exhibit a two-molecule spacing in their stoichiometry distributions.” What does this mean? Does this mean that the scaffold can increase or decrease only (mostly) by two molecules at a time? Please reformulate

Response:

For VIN3 and VRN5, we observe mostly even numbers of stoichiometry (and even intervals in distribution of stoichiometry values) when comparing different assemblies with one another; hence, we observe a two-molecule spacing in the stoichiometry distribution. The idea that assemblies would grow/decline by addition/loss of dimer units is an intuitive explanation for this observation, especially since VEL proteins undergo head-to-tail polymerisation. However, this remains a hypothesis to be tested in the future since we do not directly observe any molecular kinetics, or probe any structure, that would allow us to comment on the pathways of scaffold assembly or disassembly. We have added this important caveat to the text (lines 536-541).

4.44 “The number of labelled molecules in each assembly shows consistent periodicity of a) VRN5-YFP and b) VIN3-GFP.” Sounds circular to me (a number of molecules showing periodicity of molecules?). please reformulate.

Response:

We have now rephrased this to “The number of labelled molecules in each assembly (stoichiometry) shows consistent peak-to-peak spacing via periodicity analysis” (line 546).

4.45 “The fixed kernel width of 0.6 molecules corresponds to the rms error in the observation of a single molecule.” What is that? Where is the value of 0.6 molecules coming from? About what informs the fixed kernel? What is the rms error?

Response:

What we are aiming to detail in this section of the manuscript is the granular scale of the stoichiometry distribution; for example, if we observed a set of assemblies with an apparent stoichiometry of 9 molecules, does that distinguish them from having 10 with high confidence or not? Equivalently, if we plotted these stoichiometries as a histogram, what would be the appropriate bin width to use? To help to objectively address these questions we start by estimating the uncertainty in the observed intensities of all foci consistent which are consistent with a stoichiometry of 1, i.e. single molecules. The average of these intensities corresponds to the characteristic molecular brightness, e.g. 76 photons for one YFP molecule, with a standard error of 10 photons (equivalent to $10/76 = 13\%$ of one molecule). However, to distinguish between two observed stoichiometries, the standard deviation is a more appropriate metric since this accounts for the finite number of observations in the initial intensity of each track. The 'root mean-squared' (rms) error is another way for describing the standard deviation; for clarity we have reworded this to use simply 'standard deviation' throughout. The standard deviation of the intensities can be small when there is a low background (Supplementary Fig. 1), but in the limit of detection at low signal-to-noise, it can be as high as approximately 60% of the characteristic molecular brightness, i.e. equivalent to the brightness of 0.6 molecules. We therefore use this as a sensible and objective measure of the kernel width in the kernel density estimates of stoichiometry which circumvents potential subjective bias in using arbitrary bin widths in a histogram.

4.46 Reviewer #4 (Remarks on code availability): I have not reviewed the code functionality as this is not my expertise, but I could see on the Github link provided that the codes seem outdated (last modifications 2-6 years ago), and as mentioned in my comments to the authors, I cannot find all the scripts corresponding to the methods described in the paper - or if they are available, they are not easily findable.

Response:

We thank the reviewer for the reminder to update this repository and apologise for not doing this sooner. We now store and document all of the updated ADEMScode scripts and dependencies in the GitHub, to which we signpost each script at the relevant point in the Methods section of the manuscript.

4.47 The authors should also update the training dataset with one from SlimVar and not only from SlimField as currently on the Github space.

Response:

This is a great suggestion, thank you. We have included a minimal dataset of 10 TIFF fields of view, extracted from the VIN3-GFP data, on which all the scripts can be run and compared to exemplar output.

The manuscript by Payne-Dwyer *et al.* presents SlimVar, a new imaging technique that enables rapid single-molecule tracking of chromatin regulators in live plant cells (specifically in *Arabidopsis thaliana*). The study focuses on the epigenetic regulators VIN3 and VRN5, demonstrating that these proteins form dynamic nuclear assemblies during the vernalisation process. These assemblies increase in size and number during cold exposure, suggesting a role in maintaining epigenetic memory by stabilising gene silencing at the FLC locus. SlimVar's ability to track these processes *in vivo* offers new insights into the molecular mechanisms underlying the epigenetic regulation of flowering in plants.

Single particle tracking (SPT) on cellular membranes is a well-established technology for its compatibility with typical TIRF-based (low working distance) microscopes. Therefore, large insight into the nanoscopic properties of cell membranes has been gained over the decades. However, SPT deep within tissue (plant and other and large working distance) is a complex problem that needs addressing. The authors present this here using SlimVar to track chromatin-protein assemblies in the nuclei of single cells 30 μm deep in root tips. This is no mean feat and the data is well presented; the labs involved also have an excellent track record of high quality single-molecule data.

In our opinion the paper's primary strength lies in the development and application of the SlimVar technique, particularly the use of quantified single-molecule photobleaching at long working distances—which is innovative and of significant interest for *in vivo* studies. This method will likely drive the paper's broad appeal, particularly in journals like Nature Communications. We have some comments below that we would like the authors to address, but are very supportive of the work in general and believe it would make an excellent article in Nature communications.

Major Comments

1. **Optical Description of SlimVar:** While the manuscript introduces the new SlimVar microscopy technique, much of the critical optical description is buried in the methods section. Bringing this information forward into the main text would strengthen the manuscript and make it more accessible to a broader readership.
2. **Optical Setup Clarity:** The optical setup diagram in Figure 1 is unclear. Specifically, the depiction of the excitation beam being focused parallel to the optical axis but not collinear, alongside the collection of fluorescence across the entire back focal plane (BFP), needs improvement. Please consider redrawing the diagram or providing a more detailed version in the Supplementary Information (SI).
3. **Depth Estimation in Imaging:** The claim of imaging “30 μm deep into live tissue” in Figure 1 is based on a reference to a previous paper from the senior author, but this remains unclear (“ADEMScode”). More explicit details on how this depth is measured or confirmed is needed.
4. **Refractive Index Mismatch and Imaging Depth:** The manuscript does not adequately address the issue of refractive index mismatch at a depth of 30 μm from the coverslip, particularly when using oil immersion objectives into aqueous media. Typically, such objectives are ineffective at this working distance. The authors should explain how they overcome this challenge and discuss the implications for the effective numerical aperture (NA) and imaging depth.
5. **Depth-of-Field Clarification:** Current Figure 1 suggests a depth-of-field of 30 μm , which may be misleading. I think perhaps the authors mean “working distance” here, which is large ($\sim 30 \mu\text{m}$), whereas the focal depth is relatively low with a high NA objective ($\sim 0.5 \mu\text{m}$). Please clarify/define the actual depth-of-field in your experimental setup to avoid confusion.
6. **Tracking Metrics and Photon Quantification:** There is limited discussion of tracking metrics, such as the variation in quality with the number of points per track and detected photons. Including a histogram of track lengths would better illustrate the capabilities of SlimVar. Additionally, the manuscript currently reports values in “counts” for photon flux, which should be converted to “photons” to provide a more accurate and interpretable measure.
7. **Autofluorescence Background Quantification:** The manuscript lacks quantification of the autofluorescence background, especially deep within the roots, where it is likely to be high at the quoted power densities (kW/cm^2). A quantitative characterisation of this background, perhaps

using a wild-type control to check for false positives and absolute quantification of autofluorescence photons, would strengthen the findings.

8. **Imaging Data of Single Fluorescent Proteins (FPs):** Providing imaging data of single FPs would make the description of SlimVar more robust. Additionally, testing the approach by matching signal-to-background ratio by turning down the laser power until single molecules become difficult to detect would help demonstrate the minimum photon flux detectable by SlimVar. We would expect data showing spatially isolated puncta with single step bleaching (like those shown below) to be readily achieved.

The authors mention that “*in vitro values for characteristic single molecule brightness were obtained from E. coli recombinant fluorescent proteins at a coverslip surface under the same SlimVar imaging conditions*” but do not provide this data. Including this data, as well as histograms of photon budgets per molecule would emphasise the power of SlimVar and confirm single-molecule detection capabilities.

9. **Localisation Precision and Oligomeric Protein Assemblies:** The manuscript does not adequately address how the presence of approximately 20 molecules in oligomeric protein assemblies affects localisation precision. It is important to discuss whether the analysis code accounts for deviations from a 2D Gaussian profile in these cases.
10. **Single Molecule Imaging in Supplementary Movies:** The Supplementary Movies showing single molecules lack contrast, making it difficult to identify them in raw images, especially at long time points. Providing high-contrast grayscale versions of these movies (not in lookup table [LUT] format) would make the signal-to-noise ratio clearer and more convincing.

Minor Comments:

1. **Figure Clarity:** The figure panels, particularly 1a (optical setup) and 5a (scanning), are difficult to read and interpret. Improving the clarity and distinguishing features in these panels would make the manuscript more accessible.
2. **Colour Usage in Figures:** The colours used in Figure 2, especially in panel b, are hard to distinguish. Using complementary colours instead of different shades would improve accessibility and readability.
3. **Quantification:** Please convert all “counts” values to “photons” (*i.e* SI Fig 5 and “The characteristic single molecule brightness was 172 ± 15 (counts, mean \pm sem) for VIN3-GFP, 126 ± 16 for VRN5-YFP and LacI-YFP, 131 ± 25 for VIN3-SYFP2 and 140 ± 18 for VRN5” again these values should be measured in photons).
4. **Raw Data in Figure 3:** In Figure 3, it would be helpful to show the raw trace, including the aperiodic data points, in the non-expanded view.
5. **Dimerisation Wording on Page 8:** The wording in the section describing the confirmation that dimerisation was not an artefact of the fluorescent probe (page 8) is confusing. Rewriting this section and elaborating on the plots in SI Figure 2 would improve clarity for a broader audience.